# Spin relaxation: under the sun, anything new?[*]

Bogdan A. Rodin[1,2,3] and Daniel Abergel[1]

[1]Laboratoire des biomolécules, LBM, Département de chimie, Ecole normale supérieure, PSL University, Sorbonne Université, CNRS, 75005 Paris, France
[2]International Tomography Center, Siberian Branch of the Russian Academy of Science, Novosibirsk 630090, Russia
[3]Novosibirsk State University, Novosibirsk 630090, Russia

**Correspondence:** Daniel Abergel (daniel.abergel@ens.fr)

**Abstract.** Spin relaxation has been at the core of many studies since the early days of NMR, and the undelying theory worked out by its founding fathers. This Bloch-Redfield-Abragam relaxation theory has been recently reinvestigated (Bengs and Levitt (2020)) in the perspective of Linblad theory of quantum Markovian master equations in order to account for situations where the widely used semi-classical relaxation theory breaks down. In this article, we review the conventional approach of quantum mechanical theory of NMR relaxation and show that under the usual assumptions, it is equivalent to the Linblad formulation. We also comment on the debate over semi-classical versus quantum versions of spectral density functions involved in relaxation.

## 1 Introduction

Relaxation is the process through which a system loses energy to its environment to eventually reach a state of thermal equilibrium. Spin-lattice relaxation has been described so as to describe the way spins transfers energy to orientation degrees of freedom. Since the early days of NMR, it was the subject of numerous studies, both theoretical and encompassing a wide range of domains of applications. NMR relaxation theory has really been contemporary of the early days of NMR, and it was formalized by several of its "founding fathers" (Bloembergen et al. (1948); Bloch (1957, 1956); Abragam (1961); Redfield (1957, 1965)). This is a rather usual approach in the theory of open systems, which has been widely used in various domains of Physics (VanKampen (1981)). In this perspective, relaxation is the result of the dynamical coupling of a small ensemble of spins (the "system") coupled to a large ensemble of particles, or "degrees of freedom" (the "lattice") that is at thermal (Boltzmann) equilibrium and is endowed with an infinite heat capacity, thereby constituting a thermal reservoir. This very general approach has led to many theoretical predictions with far reaching practical applications in the domain of magnetic resonance spectroscopy. In particular, the role played by molecular motions has been put in good use to extract dynamical information on complex molecular objects. Thus, with the development of "spin engineering" techniques, it has become possible to measure selected relaxation rates with high accuracy in a broad range of problems, with the prospect of relating such observables to models of molecular dynamics.

---

[*]This paper is dedicated to Prof. Geoffrey Bodenhausen on the occasion of his 70th birthday.

It has been shown recently by Bengs and Levitt (Bengs and Levitt (2020)) that the formulation of Redfield's semiclassical theory of relaxation widely used by NMR spectroscopists may lead to erroneous predictions in the case of a two-spin system prepared in high order states, such as singlet spin states.

Such an unexpected behaviour was ascribed to the fact that some of the assumptions of the theory may not be fulfilled in NMR systems, and the authors solved the problem by making use of the Lindblad operator, which is commonly used in the theory of open quantum systems to account for dissipative Markovian phenomena, i.e., relaxation processes. Among other properties, the structure of the Lindblad operator ensures that the fundamental properties of the density operator $\rho$ ($\rho$ is hermitian and definite positive, and $\text{Tr}(\rho) = 1$) are preserved (Lindblad (1976); Alicki and Lendi (2007)). This article has sparked renewed interest regarding the general theory of relaxation, as first elaborated by Bloch, and its connection with the Linblad theory of quantum dissipative systems (Barbara (2021)). This approach has been recently generalized in several recent works (Nathan and Rudner (2020); Maimbourg et al. (2021)).

The traditional description of NMR relaxation relies on the description of both the spin system and the lattice as quantum systems, an approach that leads to the celebrated Redfield equation (Bloch (1956); Redfield (1965); Hubbard (1961); see also Goldman (2001) for a more recent account on spin lattice relaxation). As far as the description of the lattice is concerned, this approach is challenging, and actually untractable, as a quantized description of the degrees of freedom involved in molecular motions (multiple bond rotations, overall tumbling of a molecule,...) is not manageable in practice, even for small molecular systems. For this reason, an alternative relaxation theory where the spins are treated as quantum objects and the lattice is described with classical functions of the lattice degrees of freedom was developed. This semi-classical approach has far reaching practical consequences, as spin relaxation can then in principle be described using classical models of dynamics for molecular motions, and has been extensively used over the years to describe and interpret spin relaxation experiments. However, this "semi-classical" theory predicts a non-Boltzmann equilibrium density operator, which requires an *ad hoc* thermal correction to the relaxation equations (Redfield (1965); Abragam (1961)). More elaborate attempts have been made to overcome this limitation by modifying the relaxation operator itself and to enforce a Boltzman equilibrium of the density operator (Jeener (1982); Levitt and Bari (1992)). However, in the traditional semi-classical NMR relaxation approach and its later modifications, the spin-lattice interactions are accounted for by a stochastic, fluctuating, hamiltonian. This has important consequences. First, in the conventional Abragam-Redfield approach, the statistical properties of the (classical) lattice are not constrained by a fluctuation-dissipation kind of theorem that would enforce a Boltzman equilibrium distribution, therefore a lattice temperature. Such a constraint is therefore not applied on the spins. Secondly, a major difference between the quantum and classical theories of spin relaxation is rooted in the non commutation of quantum bath operators of the spin-lattice coupling hamiltonian, which confers particular properties to the spin correlation and spectral density functions that are absent in the semi-classical theory. Both aspects, quantum and statistical, are entangled, and the relations between quantum and classical correlation functions will be therefore discussed.

It is the purpose of this paper to re-investigate these old questions in order to trace the roles and the consequences of the various assumptions of the traditional approach to relaxation developed in the early days of NMR.

## 2 Theory of spin relaxation in a thermal bath: a short review

### 2.1 Derivation of the master equation

The derivation follows the lines of Refs. (Abragam (1961); Bloch (1956); Redfield (1957); Hubbard (1961)). Consider a spin system in its environment. The hamiltonian of the spin system $H_S$ accounts for the interaction of the spins with the magnetic fields (Zeeman interaction and interaction with a rf field), as well as non dissipative spin-spin interactions (scalar J coupling, dipolar coupling). The dynamics of the lattice (the bath) is described by $H_B$, and the spin-bath interaction hamiltonian is $H_1$:

$$H_T = H_S + H_B + H_1 \tag{1}$$

Thus, the dynamics of the {spin-bath} system is described by the Liouville equation:

$$\dot{\rho}(t) = -i[H_S + H_B + H_1, \rho(t)] \tag{2}$$

Alternatively, it can be written as:

$$\dot{\rho}(t) = \mathcal{L}_T \rho(t) \tag{3}$$

where the operator $\mathcal{L}_T = -i[H_T, \cdot]$ is the total Liouvillian. Introducing the interaction representation of the density operator $\rho^*(t) = e^{-\mathcal{L}_0 t}\rho(t)$, where $\mathcal{L}_0 = \mathcal{L}_S + \mathcal{L}_B$ is the unperturbed liouvillian, one has:

$$\frac{d}{dt}\rho^*(t) = -i[H_1^*(t), \rho^*(t)] = \mathcal{L}_1^*(t)\rho^*(t) \tag{4}$$

The Liouville equation is integrated to second order:

$$\rho^*(t) = \rho^*(0) + \int_0^t dt_1 \mathcal{L}_1^*(t_1)\rho^*(0) + \int_0^t dt_1 \int_0^{t_1} dt_2 \mathcal{L}_1^*(t_1)\mathcal{L}_1^*(t_2)\rho^*(0) + \ldots \tag{5}$$

Taking the derivative, one gets:

$$\frac{d}{dt}\rho^*(t) = \mathcal{L}_1^*(t)\rho^*(0) + \int_0^t dt' \mathcal{L}_1^*(t)\mathcal{L}_1^*(t')\rho^*(0) \tag{6}$$

Finally, by making the change of variables $\tau = t - t'$, one gets the master equation for the total density operator as:

$$\frac{d}{dt}\rho^*(t) = \mathcal{L}_1^*(t)\rho^*(0) + \int_0^t d\tau \mathcal{L}_1^*(t)\mathcal{L}_1^*(t-\tau)\rho^*(0) \tag{7}$$

The dynamics restricted to the spin system is obtained by eliminating the bath variables. This is achieved by performing a partial trace over the bath degrees of freedom:

$$\sigma(t) = \text{tr}_B\left\{\rho(t)\right\} = \text{tr}_B\left\{e^{-iH_T t}\rho(0)e^{iH_T t}\right\} = \text{tr}_B\left\{e^{\mathcal{L}_T t}\rho(0)\right\} \tag{8}$$

Hence, from Eq. 6, the spin density operator in the interaction representation:

$$\sigma^*(t) = \text{tr}_{\text{B}} \rho^*(t) \tag{9}$$

obeys:

$$\frac{d}{dt}\sigma^*(t) = \text{tr}_{\text{B}} \left\{ \mathcal{L}_1^*(t_1)\rho^*(0) \right\} + \int_0^t dt' \text{tr}_{\text{B}} \left\{ \mathcal{L}_1^*(t)\mathcal{L}_1^*(t')\rho^*(0) \right\} \tag{10}$$

Initially, the system and the bath are assumed to be completely decorrelated:

$$\rho(0) = \rho_{\text{B}}^e \otimes \sigma(0) \tag{11}$$

where, the exact $\rho_{\text{B}}^e$ denotes the bath density operator in thermal equilibrium. With these assumptions, one has:

$$\frac{d}{dt}\sigma^*(t) = \text{tr}_{\text{B}} \left\{ \mathcal{L}_1^*(t_1)\rho^*(0) \right\} + \int_0^t dt' \text{tr}_{\text{B}} \left\{ \mathcal{L}_1^*(t)\mathcal{L}_1^*(t')\rho_{\text{B}}^e \right\} \sigma^*(0) \tag{12}$$

In the latter expression, the term $\text{tr}_{\text{B}} \left\{ \mathcal{L}_1^*(t_1)\rho^*(0) \right\}$ may be assumed to be zero or can be incorporated in the main system
liouvillian $\mathcal{L}_{\text{s}}$ (Abragam (1961); Redfield (1957)):

$$\frac{d}{dt}\sigma^*(t) = \int_0^t dt' \text{tr}_{\text{B}} \left\{ \mathcal{L}_1^*(t)\mathcal{L}_1^*(t')\rho_{\text{B}}^e \right\} \sigma^*(0) \tag{13}$$

If the spin density operator is assumed to only moderately depart from its initial state,

$$\frac{\sigma(t) - \sigma(0)}{\sigma(0)} \ll 1, \tag{14}$$

the density operator varies only slightly from its initial state, so that $\sigma^*(0)$ can be replaced by $\sigma^*(t)$ in Eq. 12 (Abragam (1961);
Redfield (1957)):

$$\frac{d}{dt}\sigma^*(t) = \int_0^t dt' \text{tr}_{\text{B}} \left\{ \mathcal{L}_1^*(t)\mathcal{L}_1^*(t')\rho_{\text{B}}^e \right\} \sigma^*(t) \tag{15}$$

The master equation in the Schrödinger representation can be obtained from Eq.15:

$$\frac{d}{dt}\sigma^*(t) = \frac{d}{dt}e^{-\mathcal{L}_{\text{s}}t}\sigma(t) = -\mathcal{L}_{\text{s}}e^{-\mathcal{L}_{\text{s}}t}\sigma(t) + e^{-\mathcal{L}_{\text{s}}t}\frac{d}{dt}\sigma(t) \tag{16}$$

$$= \int_0^t dt' \text{tr}_{\text{B}} \left\{ \mathcal{L}_1^*(t)\mathcal{L}_1^*(t')\rho_{\text{B}}^e \right\} \sigma^*(t) \tag{17}$$

Therefore, one has:

$$\frac{d}{dt}\sigma(t) = \mathcal{L}_{\text{s}}\sigma(t) + e^{\mathcal{L}_{\text{s}}t}\int_0^t dt' \text{tr}_{\text{B}} \left\{ \mathcal{L}_1^*(t)\mathcal{L}_1^*(t')\rho_{\text{B}}^e \right\} \sigma^*(t) \tag{18}$$

Reverting to the Schrödinger representation $\sigma^*(t) = e^{-tL_s}\sigma(t)$, one obtains the master equation in the Schrödinger representation:

$$\frac{d}{dt}\sigma(t) = \mathcal{L}_s\sigma(t) + \int_0^t dt' \text{tr}_\text{B} \left\{ \mathcal{L}_1 \mathcal{L}_1^*(t' - t)\rho_\text{B}^e \right\} \sigma(t) \tag{19}$$

A derivation is given in Appendix A for reference. The term $\text{tr}_\text{B} \left\{ \mathcal{L}_1 \mathcal{L}_1^*(t' - t)\rho_\text{B}^e \right\}$ in equation 19 is a correlation operator acting on the spin system. It projects the system-bath (spin-lattice) coupling onto the bath degrees of freedom. This spin operator therefore carries the statistical properties of the bath, described by its equilibrium, stationary, density operator. Finally, assuming that the correlation operator decays to zero in a time $\tau_c$ much shorter than the period over which the density matrix varies significantly, the upper limit of the integral can be extended to $+\infty$. As above, making the change of variables $\tau = t - t'$,
one obtains:

$$\frac{d}{dt}\sigma(t) = \mathcal{L}_s\sigma(t) + \int_0^{+\infty} d\tau \text{tr}_\text{B} \left\{ \mathcal{L}_1 \mathcal{L}_1^*(-\tau)\rho_\text{B}^e \right\} \sigma(t) \tag{20}$$

or, in the hamiltonian representation:

$$\frac{d}{dt}\sigma(t) = -i[H_s, \sigma(t)] - \int_0^{+\infty} d\tau \text{tr}_\text{B} \left\{ [H_1, [H_1^*(-\tau), \rho_\text{B}^e \sigma(t)]] \right\} \tag{21}$$

One therefore obtains a master equation of the Redfield kind:

$$\frac{d}{dt}\sigma(t) = -i[H_s, \sigma(t)] + \mathcal{R}\sigma(t) \tag{22}$$

where $\mathcal{R} \bullet = - \int_0^{+\infty} d\tau \text{tr}_\text{B} \left\{ [H_1, [H_1^*(-\tau), \rho_\text{B}^e \bullet]] \right\}$ is the Redfield (relaxation) operator.

## 2.2 Formulation of the master equation in operator form

In spin relaxation theory, it is customary to express the relaxation equation in operator form, which often provides a clearer representation of the spin-bath coupling dynamics. Here, the coupling Hamiltonian is assumed to have the form of a sum of
120 terms, each of which factorizes into a product of lattice $B^q$ and spin $S^q$ operators.

$$H_1 = \sum_q S^q B^q, \tag{23}$$

with the interaction representation:

$$B^q(t) = e^{iH_\text{B}t} B^q e^{-iH_\text{B}t} \tag{24}$$
$$S^q(t) = e^{iH_s t} S^q e^{-iH_s t} \tag{25}$$

Using results of the preceding section (Eqs 16-19), the Redfield equation becomes:

$$\dot{\sigma}_{\mathrm{S}}^*(t) = -\sum_{q,q'} \mathrm{tr}_{\mathrm{B}} \int_0^{+\infty} dt' [S^q(t)B^q(t), [S^{q'}(t')B^{q'}(t'), \rho_B^e \sigma_{\mathrm{S}}^*(t)]] \tag{26}$$

Each term in the sum becomes:

$$\mathrm{tr}_{\mathrm{B}}[S^q(t)B^q(t), [S^{q'}(')B^{q'}(t'), \rho_B^e \sigma_{\mathrm{S}}^*(t)]] =$$
$$[S^q(t), S^{q'}(t')\sigma_{\mathrm{S}}^*(t)]\langle B^q(t)B^{q'}(t')\rangle^e + [\sigma_{\mathrm{S}}^*(t)S^{q'}(t'), S^q(t)]\langle B^{q'}(t')B^q(t)\rangle^e \tag{27}$$

where the notation:

$$\langle B^q(t)B^{q'}(t')\rangle^e = \mathrm{tr}_{\mathrm{B}}\left\{ B^q(t)B^{q'}(t')\rho_B^e \right\} \tag{28}$$

has been introduced. The $\langle B^q(t)B^{q'}(t')\rangle^e$ are the bath (lattice) correlation functions, and in contrast to equation 20, these denote usual time correlation functions rather than operators.

$$\dot{\sigma}_{\mathrm{S}}^*(t) = -\sum_{q,q'} \int_0^{+\infty} dt' [S^q(t), S^{q'}(t')\sigma_{\mathrm{S}}^*(t)]\langle B^q(t)B^{q'}(t')\rangle^e$$
$$- \sum_{q,q'} \int_0^{+\infty} dt' [\sigma_{\mathrm{S}}^*(t)S^{q'}(t'), S^q(t)]\langle B^{q'}(t')B^q(t)\rangle^e \tag{29}$$

Using the conventional decomposition of the spin operators into a sum of eigenoperators of the liouvillian $\mathcal{L}_{\mathrm{S}} = [H_{\mathrm{S}}, \bullet]$:

$$[H_{\mathrm{S}}, S_p^q] = \omega_p^q S_p^q \tag{30}$$

one has:

$$S^q(t) = e^{iH_{\mathrm{S}}t}S^q e^{-iH_{\mathrm{S}}t} = \sum_p S_p^q e^{i\omega_p^q t} \tag{31}$$

one obtains from equation 29, with the change of integration variable $\tau = t - t'$:

$$\dot{\sigma}_{\mathrm{S}}^*(t) = -\sum_{q,q',p,p'} e^{i(\omega_p^q + \omega_{p'}^{q'})t} \int_0^{+\infty} d\tau [S_p^q, S_{p'}^{q'}\sigma_{\mathrm{S}}^*(t)]\langle B^q(t)B^{q'}(t-\tau)\rangle^e e^{-i\omega_{p'}^{q'}\tau}$$
$$- \sum_{q,q',p,p'} e^{i(\omega_p^q + \omega_{p'}^{q'})t} \int_0^{+\infty} d\tau [\sigma_{\mathrm{S}}^*(t)S_{p'}^{q'}, S_p^q]\langle B^{q'}(t-\tau)B^q(t)\rangle^e e^{-i\omega_{p'}^{q'}\tau} \tag{32}$$

Intoducing the secular approximation $\omega_p^q + \omega_{p'}^{q'} = 0$, so that $p = p', q = -q'$, and renaming indices, this reduces to:

$$\dot{\sigma}_{\mathrm{S}}^*(t) = -\sum_{p,q} \int_0^{+\infty} d\tau [S_p^{-q}, S_p^q \sigma_{\mathrm{S}}^*(t)]\langle B^{-q}(t)B^q(t-\tau)\rangle^e e^{-i\omega_p^q \tau}$$
$$- \sum_{p,q} \int_0^{+\infty} d\tau [\sigma_{\mathrm{S}}^*(t)S_p^q, S_p^{-q}]\langle B^q(t-\tau)B^{-q}(t)\rangle^e e^{-i\omega_p^q \tau} \tag{33}$$

The assumption that the bath is in a stationary state, $[H_\text{B}, \rho_{B,}^e] = 0$, confers some properties to the correlation functions. Thus, the bath correlation functions are also stationary. Indeed, one has:

$$
\begin{aligned}
\langle B^q(t)B^{-q}(t+\tau)\rangle^e &= \text{tr}_\text{B}\{e^{iH_\text{B}t}B^q e^{-iH_\text{B}t}e^{iH_\text{B}(t+\tau)}B^{-q}e^{-iH_\text{B}(t+\tau)}\rho_B^e\} \\
&= \text{tr}_\text{B}\{e^{iH_\text{B}(t-\tau)}B^q e^{-iH_\text{B}(t-\tau)}e^{iH_\text{B}t}B^{-q}e^{-iH_\text{B}t}\rho_B^e\} \\
&= \langle B^q(t-\tau)B^{-q}(t)\rangle^e
\end{aligned}
\tag{34}
$$

In addition, because $\text{tr}(AB)^* = \text{tr}(B^\dagger A^\dagger)$, it is easy to show that:

$$
\langle B^q(t-\tau)B^{-q}(t)\rangle^{e*} = \langle B^q(t)B^{-q}(t-\tau)\rangle^e
\tag{35}
$$

Besides, using the property that $B^{-q} = B^{q\dagger}$:

$$
\begin{aligned}
\langle B^{-q}(t)B^q(t-\tau)\rangle^e &= \frac{1}{L}\sum_{f,f'}\langle f|e^{iH_\text{B}t}B^{-q}e^{-iH_\text{B}t}|f'\rangle\langle f'|e^{iH_\text{B}(t-\tau)}B^q e^{-iH_\text{B}(t-\tau)}e^{-\beta H_\text{B}}|f\rangle \\
&= \frac{1}{L}\sum_{f,f'}\langle f|B^{-q}|f'\rangle e^{ift}e^{-if't}\langle f'|B^q|f\rangle e^{if'(t-\tau)}e^{-if(t-\tau)}e^{-\beta f} \\
&= \frac{1}{L}\sum_{f,f'}\langle f|B^{-q}|f'\rangle\langle f'|B^q|f\rangle e^{-i(f'-f)\tau}e^{-\beta f} \\
&= \frac{1}{L}\sum_{f,f'}|\langle f'|B^q|f\rangle|^2 e^{-i(f'-f)\tau}e^{-\beta f}
\end{aligned}
\tag{36}
$$

where the $|f\rangle$ are the eigenstates of the bath hamiltonian $H_\text{B}$, and $\beta = \frac{\hbar}{kT}$, with the convention $\hbar = 1$. In these equation, the notation $L = \text{tr}_\text{B}e^{\beta H_\text{B}}$ was introduced.

## 3   The Redfield equation is equivalent to the Lindblad form of the relaxation equation

It is now straightforward to show that the conventional Bloch-Redfield-Abragam perturbative approach of relaxation is equivalent to the Lindblad formulation of dissipative systems. Indeed, changing indices in the first term and using the property $\omega_p^{-q} = -\omega_p^q$), and setting $\tau \to -\tau$ one has, from Eq. 33:

$$
\begin{aligned}
\dot{\sigma}_\text{S}^*(t) &= -\sum_{p,q}\int_{-\infty}^{0} d\tau[S_p^q, S_p^{-q}\sigma_\text{S}^*(t)]\langle B^q(t)B^{-q}(t+\tau)\rangle^e e^{-i\omega_p^q\tau} \\
&\quad - \sum_{p,q}\int_{0}^{+\infty} d\tau[\sigma_\text{S}^*(t)S_p^q, S_p^{-q}]\langle B^q(t-\tau)B^{-q}(t)\rangle^e e^{-i\omega_p^q\tau}
\end{aligned}
\tag{37}
$$

Using the stationarity property (equation 34) of the bath correlation functions leads to:

$$
\begin{aligned}
\dot{\sigma}_{\mathrm{S}}^*(t) &= -\sum_{p,q} \int_{-\infty}^{0} d\tau [S_p^q, S_p^{-q}\sigma_{\mathrm{S}}^*(t)] \langle B^q(t-\tau)B^{-q}(t)\rangle^e e^{-i\omega_p^q \tau} \\
&\quad - \sum_{p,q} \int_{0}^{+\infty} d\tau [\sigma_{\mathrm{S}}^*(t)S_p^q, S_p^{-q}] \langle B^q(t-\tau)B^{-q}(t)\rangle^e e^{-i\omega_p^q \tau}
\end{aligned}
\tag{38}
$$

<p>170</p>

$$
\begin{aligned}
\dot{\sigma}_{\mathrm{S}}^*(t) &\approx -\frac{1}{2}\sum_{p,q} \int_{-\infty}^{+\infty} d\tau [S_p^q, S_p^{-q}\sigma_{\mathrm{S}}^*(t)] \langle B^q(t-\tau)B^{-q}(t)\rangle^e e^{-i\omega_p^q \tau} \\
&\quad - \frac{1}{2}\sum_{p,q} \int_{-\infty}^{+\infty} d\tau [\sigma_{\mathrm{S}}^*(t)S_p^q, S_p^{-q}] \langle B^q(t-\tau)B^{-q}(t)\rangle^e e^{-i\omega_p^q \tau}
\end{aligned}
\tag{39}
$$

so that:

$$
\begin{aligned}
\dot{\sigma}_{\mathrm{S}}^*(t) &\approx -\frac{1}{2}\sum_{p,q}[S_p^q, S_p^{-q}\sigma_{\mathrm{S}}^*(t)]J_{\mathrm{R}}^{q,-q}(\omega_p^q) - \frac{1}{2}\sum_{p,q}[\sigma_{\mathrm{S}}^*(t)S_p^q, S_p^{-q}]J_{\mathrm{R}}^{q,-q}(\omega_p^q) \\
&= \sum_{p,q} J_{\mathrm{R}}^{q,-q}(\omega_p^q)\left(S_p^{-q}\sigma_{\mathrm{S}}^*(t)S_p^q - \frac{1}{2}\left\{S_p^q S_p^{-q}, \sigma_{\mathrm{S}}^*(t)\right\}\right)
\end{aligned}
\tag{40}
$$

<p>175</p> where the "right" spectral density $J_{\mathrm{R}}^{q,-q}(\omega_p^q)$ of the bath is given by:

$$
J_{\mathrm{R}}^{q,-q}(\omega_p^q) = \int_{-\infty}^{+\infty} d\tau \langle B^q B^{-q}(\tau)\rangle e^{-i\omega_p^q \tau} = \int_{-\infty}^{+\infty} d\tau C_{\mathrm{R}}^{q,-q}(\tau) e^{-i\omega_p^q \tau}
\tag{41}
$$

$\{\cdot,\cdot\}$ denotes the anti-commutator. The operator appearing in the right handside of Eq. 40 is the Lindblad dissipation superoperator as:

$$
\hat{\mathscr{L}\mathscr{D}}[A, B] = A \bullet B - \frac{1}{2}\left(BA \bullet + \bullet BA\right),
\tag{42}
$$

<p>180</p> so that equation 40 becomes:

$$
\dot{\sigma}_{\mathrm{S}}^*(t) = \sum_{p,q} J_R^{q,-q}(\omega_p^q)\hat{\mathscr{L}\mathscr{D}}[S_p^{-q}, S_p^q]\sigma_{\mathrm{S}}^*(t).
\tag{43}
$$

Equation 43 (as well as 40) is a Lindblad equation (Lindblad (1976); Alicki and Lendi (2007)), which is thus derived from the usual quantized theory of relaxation (Bloch (1957); Redfield (1957); Hubbard (1961); Abragam (1961)). The fact that this derivation leads to the Lindblad equation is not obvious. In principle, one should not expect the perturbative approach to yield
<p>185</p> an irreversible dissipative operator equivalent to a Lindblad operator. In fact, this equivalence requires the Markovian and the short correlation time assumptions that make the evolution equation depend on the density operator at the present time t only, not on its previous history. Moreover, it also requires the secular approximation that eliminates the time dependence of the spin

operators of the coupling hamiltonian in Eq. 29, leading to Eq. 33. The combination of these conditions lead to the semi-group property and the Lindblad form of the relaxation operator.

Another point is worth mentioning. The properties of the correlation functions that emerge through this procedure reflect properties of the bath operators of the spin-bath coupling hamiltonian, and therefore convey additional properties that are not implied by the structure of the Lindblad equations 40 and 43. Such properties, arising from non commutation of the bath operators and of the fact that the lattice is always in stationary Boltzmann equilibrium, and detailed in section 3.1 below and in Appendix B, ensure detailed balance and that the stationary state of the spin system is compatible with the Boltzmann

equilibrium of the lattice, that is, the lattice temperature. In other words, the perturbative approach leads to a master equation of the Lindblad form, with additional physical properties that bear constraints from the lattice.

### 3.1    An alternative formulation (equivalent to Lindblad)

It is possible to obtain an alternative and completely equivalent form of the Redfield equation. Expanding Eq. 33, one gets:

$$
\begin{aligned}
\dot{\sigma}_S^*(t) &= -\sum_q \int_0^\infty d\tau\, S_p^{-q} S_p^q \sigma_S^*(t) \langle B^{-q}(\tau) B^q \rangle e^{-i\omega_p^q \tau} \\
&+ \sum_q \int_0^\infty d\tau\, S_p^{-q} \sigma_S^*(t) S_p^q \langle B^q B^{-q}(\tau) \rangle e^{-i\omega_p^q \tau} \\
&+ \sum_q \int_0^\infty d\tau\, S_p^q \sigma_S^*(t) S_p^{-q} \langle B^{-q}(\tau) B^q \rangle e^{-i\omega_p^q \tau} \\
&- \sum_q \int_0^\infty d\tau\, \sigma_S^*(t) S_p^q S_p^{-q} \langle B^q B^{-q}(\tau) \rangle e^{-i\omega_p^q \tau}
\end{aligned}
\tag{44}
$$

As above, the "left-sided" spectral density function is defined as:

$$
J_{\mathrm{L}}^{-q,q}(\omega) = \int_{-\infty}^\infty d\tau \langle B^{-q}(\tau) B^q \rangle e^{-i\omega\tau} = \int_{-\infty}^\infty d\tau\, C_{\mathrm{L}}^{-q,q}(\tau) e^{-i\omega\tau} = e^{-\beta\omega} J_{\mathrm{R}}^{q,-q}(\omega)
\tag{45}
$$

where $\beta = \hbar/kT$ ($\hbar = 1$). The latter equations express a Kubo kind of relation (Kubo (1957)): $J_{\mathrm{L}}^{-q,q}(\omega) = e^{-\beta\omega} J_{\mathrm{R}}^{q,-q}(\omega)$. A proof is given in the Appendix (see Eqs B1 and B2). After some straightforward manipulations one obtains:

$$
\begin{aligned}
\dot{\sigma}_S^*(t) &= -\frac{1}{2} \sum_q S_p^{-q} S_p^q \sigma_S^*(t) e^{-\beta\omega_p^q} J_{\mathrm{R}}^{q,-q}(\omega_p^q) + \frac{1}{2} \sum_q S_p^{-q} \sigma_S^*(t) S_p^q J_{\mathrm{R}}^{q,-q}(\omega_p^q) \\
&+ \frac{1}{2} \sum_q S_p^q \sigma_S^*(t) S_p^{-q} e^{-\beta\omega_p^q} J_{\mathrm{R}}^{q,-q}(\omega_p^q) - \frac{1}{2} \sum_q \sigma_S^*(t) S_p^q S_p^{-q} J_{\mathrm{R}}^{q,-q}(\omega_p^q)
\end{aligned}
\tag{46}
$$

Thus, collecting and rearranging terms, one gets:

$$
\dot{\sigma}_S^*(t) = \frac{1}{2} \sum_q J_{\mathrm{R}}^{q,-q}(\omega_p^q) \left( [S_p^{-q}, \sigma_S^*(t) S_p^q] - [S_p^{-q}, S_p^q \sigma_S^*(t)] e^{-\beta\omega_p^q} \right)
\tag{47}
$$

The Boltzmann factor can be expanded in series:

$$\dot{\sigma}_{\text{S}}^*(t) = -\frac{1}{2}\sum_q J_{\text{R}}^{q,-q}(\omega_p^q)[S_p^{-q},[S_p^q,\sigma_{\text{S}}^*(t)]] - \frac{1}{2}\sum_q J_{\text{R}}^{q,-q}(\omega_p^q)[S_p^{-q},S_p^q\sigma_{\text{S}}^*(t)\sum_{n=1}^{\infty}\frac{1}{n!}(-\beta\omega_p^q)^n] \tag{48}$$

The first term on the right hand side of this equation is the usual double commutator, and the symmetry while the second term represents the thermal effect of the Boltzmann equilibrium of the lattice. This term, equal to $1 - \exp\left(-\beta\omega_p^q\right)$, vanishes for infinite temperature.

## 4   A Pseudo-classical version of the Redfield equation

A semi-classical version of the master equation can be extremely useful, allowing one to make use of models derived in the framework of classical mechanics to calculate spectral density functions. In order to obtain such a theory associated to Eqs. 40 and 43, additional adjustments are necessary. Indeed, because the $B^{-q}(\tau)$ operators do not commute, the correlation functions of the type $\langle B^{-q}(\tau)B^q \rangle$ do not obey the general symmetry rules of classical correlation functions. However, symmetrized correlation functions do commute and so these symmetrized (quantum-mechanical) correlation functions should be introduced in order to obtain a semi-classical theory (which we may call « pseudo-classical » to distinguish it from the theory where the effect of the bath is taken into account only through random functions). A general definition of the classical correlation function of two dynamical variables $\mathbf{A}$ and $\mathbf{B}$ is (Evans and Moriss (2008)):

$$C_{\mathbf{AB}}(t) = \langle \mathbf{A}(t)\mathbf{B}^* \rangle \tag{49}$$

where the brackets indicate classical ensemble average. In the case of stationary processes, the following properties of a correlation functions can be deduced. Its complex conjugate $C_{\mathbf{AB}}^*(t)$ is therefore (Evans and Moriss (2008)):

$$C_{\mathbf{AB}}^*(t) = \langle \mathbf{A}(t)\mathbf{B}^* \rangle^* = \langle \mathbf{A}^*(t)\mathbf{B} \rangle = \langle \mathbf{A}^*\mathbf{B}(-t) \rangle = C_{\mathbf{BA}}(-t) \tag{50}$$

For an autocorrelation function of $\mathbf{A}$, $C_{\mathbf{AA}}(t)$, one has:

$$C_{\mathbf{AA}}(t)^* = C_{\mathbf{AA}}(-t). \tag{51}$$

Eq. 51 shows that, in the general case where the autocorrelation function is complex: $C_{\mathbf{AA}}(t) = C_{\mathbf{AA}}^r(t) + iC_{\mathbf{AA}}^i(t)$, with $C_{\mathbf{AA}}^r(t) = \mathcal{R}e(C_{\mathbf{AA}}(t))$ and $C_{\mathbf{AA}}^i(t) = \mathcal{I}m(C_{\mathbf{AA}}(t))$ are even and odd functions of time, since $C_{\mathbf{AA}}^*(t) = C_{\mathbf{AA}}^r(t) - iC_{\mathbf{AA}}^i(t) = C_{\mathbf{AA}}^r(-t) + iC_{\mathbf{AA}}^i(-t)$. This implies that the associated spectral density, $J(\omega) = \int_{-\infty}^{+\infty} C_{\mathbf{AA}}(t)e^{-i\omega t}dt$ is real and $J(\omega) = J_{\mathbf{AA}}^e(\omega) + J_{\mathbf{AA}}^o(\omega)$ $J_{\mathbf{AA}}^e(\omega) = \int_{-\infty}^{+\infty} C_{\mathbf{AA}}^r(t)e^{-i\omega t}dt$ $J_{\mathbf{AA}}^o(\omega) = i\int_{-\infty}^{+\infty} C_{\mathbf{AA}}^i(t)e^{-i\omega t}dt$ are real, and respectively even and odd functions.

A semi-classical relaxation theory should provide spectral density functions obeying the general classical mechanics requirements detailed above. It is clear, however, that in the quantum case, where $\mathbf{A}$ and $\mathbf{B}$ are in general non commuting operators, the above symmetry relations do no apply. It is nevertheless possible to define a symmetrized correlation function as:

$$C_{\mathbf{AB}}(t) = \frac{1}{2}\left\{\langle \mathbf{A}^{\dagger}\mathbf{B}(t) \rangle + \langle \mathbf{B}^{\dagger}\mathbf{A}(-t) \rangle\right\} \tag{52}$$

which is real (when $C_{\mathbf{AB}}(t)$ is stationary). Note also that the bath operator correlation functions have the property:

$$
\begin{aligned}
\langle B^q B^{q'}(\tau)\rangle^* &= \text{tr}\{B^q e^{iH_{\text{B}}\tau}B^{q'}e^{-iH_{\text{B}}\tau}\rho^e\}^* = \text{tr}\{\rho^e e^{iH_{\text{B}}\tau}B^{q'\dagger}e^{-iH_{\text{B}}\tau}B^{q\dagger}\} \\
&= \langle B^{q'\dagger}(\tau)B^{q\dagger}\rangle = \langle B^{q'\dagger}B^{q\dagger}(-\tau)\rangle
\end{aligned}
\tag{53}
$$

(correlation functions are assumed stationary), so that:

$$
\langle B^q B^{-q}(\tau)\rangle^* = \langle B^q(\tau)B^{-q}\rangle = \langle B^q B^{-q}(-\tau)\rangle
\tag{54}
$$

$$
\langle B^{-q} B^q(\tau)\rangle^* = \langle B^{-q}(\tau)B^q\rangle = \langle B^{-q}B^q(-\tau)\rangle
\tag{55}
$$

Using the definitions $C_{\text{L}}^{-q,q}(\tau) = \langle B^{-q}(\tau)B^q\rangle$ and $C_{\text{R}}^{q,-q}(\tau) = \langle B^q B^{-q}(\tau)\rangle$, the relations Eqs. 54 and 55 show that the average $C^q(\tau) = \frac{1}{2}(C_{\text{L}}^{-q,q}(\tau) + C_{\text{R}}^{q,-q}(\tau))$ obey the classical correlation function property $C^{q*}(\tau) = C^q(-\tau)$.

A semi-classical version of the Redfield equation is thus obtained by using the spectral density function $J^q(\omega)$ obtained from the Fourier transform of the symmetrized correlation function $C^q(\tau)$:

$$
J^q(\omega) = \frac{1}{2}(J_{\text{R}}^{q,-q}(\omega) + J_{\text{L}}^{-q,q}(\omega)) = \frac{1}{2}(J_{\text{R}}^{q,-q}(\omega) + e^{-\beta\omega}J_{\text{R}}^{q,-q}(\omega)) = \frac{1 + e^{-\beta\omega}}{2}J_{\text{R}}^{q,-q}(\omega)
\tag{56}
$$

Using Eqs. 56 and 46, it is therefore possible to derive an alternative expression of the master equation. This gives:

$$
\begin{aligned}
\dot{\sigma}_{\text{S}}^*(t) &= -\sum_q S_p^{-q}S_p^q\sigma_{\text{S}}^*(t)\frac{e^{-\beta\omega_p^q}}{1+e^{-\beta\omega_p^q}}J^q(\omega_p^q) + \sum_q S_p^{-q}\sigma_{\text{S}}^*(t)S_p^q\frac{1}{1+e^{-\beta\omega_p^q}}J^q(\omega_p^q) \\
&\quad + \sum_q S^q\sigma_{\text{S}}^*(t)S_p^{-q}\frac{e^{-\beta\omega_p^q}}{1+e^{-\beta\omega_p^q}}J^q(\omega_p^q) - \sum_q \sigma_{\text{S}}^*(t)S_p^q S_p^{-q}\frac{1}{1+e^{-\beta\omega_p^q}}J^q(\omega_p^q)
\end{aligned}
\tag{57}
$$

or:

$$
\begin{aligned}
\dot{\sigma}_{\text{S}}^*(t) &= -\sum_q S_p^{-q}S_p^q\sigma_{\text{S}}^*(t)\frac{1}{1+e^{\beta\omega_p^q}}J^q(\omega_p^q) + \sum_q S_p^{-q}\sigma_{\text{S}}^*(t)S_p^q\frac{1}{1+e^{-\beta\omega_p^q}}J^q(\omega_p^q) \\
&\quad + \sum_q S_p^q\sigma_{\text{S}}^*(t)S_p^{-q}\frac{1}{1+e^{\beta\omega_p^q}}J^q(\omega_p^q) - \sum_q \sigma_{\text{S}}^*(t)S_p^q S_p^{-q}\frac{1}{1+e^{-\beta\omega_p^q}}J^q(\omega_p^q)
\end{aligned}
\tag{58}
$$

Finally, collecting and rearranging terms, one gets:

$$
\dot{\sigma}_{\text{S}}^*(t) = \frac{1}{2}\sum_q J^q(\omega_p^q)\left([S_p^{-q},\sigma_{\text{S}}^*(t)S_p^q]\frac{1}{1+e^{-\beta\omega_p^q}} - [S_p^{-q},S_p^q\sigma_{\text{S}}^*(t)]\frac{1}{1+e^{\beta\omega_p^q}}\right)
\tag{59}
$$

In view of clarifying the connection between the derivation of the quantummechanical master equation to further semi-classical
approximations, it is interesting to rewrite Eq. 59 by expanding the temperature function in terms of the parameters $\beta\omega_p^q$:

$$
\begin{aligned}
\dot{\sigma}_{\text{S}}^*(t) &= \frac{1}{2}\sum_q J^q(\omega_p^q)\left([S_p^{-q},\sigma_{\text{S}}^*(t)S_p^q](\frac{1}{2}+\frac{\beta\omega_p^q}{4}-\frac{(\beta\omega_p^q)^3}{48}) - [S_p^{-q},S_p^q\sigma_{\text{S}}^*(t)](\frac{1}{2}-\frac{\beta\omega_p^q}{4}+\frac{(\beta\omega_p^q)^3}{48})\right) \\
&= \frac{1}{4}\sum_q J^q(\omega_p^q)[S_p^{-q},[S_p^q,\sigma_{\text{S}}^*(t)]] + \frac{1}{2}\sum_q J^q(\omega_p^q)(\frac{\beta\omega_p^q}{4}-\frac{(\beta\omega_p^q)^3}{48}+\cdots)[S_p^{-q},\{S_p^q,\sigma_{\text{S}}^*(t)\}]
\end{aligned}
\tag{60}
$$

Eq. 60 contains a double commutator term weighted by the spectral densities $J^q(\omega_p^q)$, constructed so as to obey the general symmetry properties of classical spectral density functions (see Eq. 49 and below) and are therefore adapted to a semiclassical version of the relaxation master equation. Then, the $J^q(\omega)$ are obtained from classical lattice functions of a fluctuating hamiltonian, whilst the semi-classical ME obeys detailed balance. The second term of Eq. 60 introduces a lattice-temperature dependent contribution. However, this term vanishes when the bath operators of the spin-bath coupling hamiltonian commute, $[B^{-q}, B^q] = 0$. According to Eq. B2, the latter condition also implies that one has the equality $J_{\rm L}^{-q,q}(\omega) = J_{\rm R}^{q,-q}(\omega)$, meaning that the lattice temperature is infinite. Stated otherwise, this means that a finite lattice temperature is incompatible with commuting bath operators. However, in general, $[B^{-q}, B^q] \neq 0$, so that detailed balance assumption, or property, which is ensured by the model of a bath in thermal Boltzmann equilibrium, is conveyed to the spin system through non commutation of the bath operators $B^q$. Each term in the series expansion in the righthand side of Eq. 60 explicitly gives the effect of non commutation, at each order of the parameter $\beta\omega_p^q$. The first order approximation provides the adequate expression in the high-temperature limit (see below).

The final relaxation superoperator, which defines the relaxation of density matrix as $\dot{\sigma}_{\rm S}^*(t) = \hat{\Gamma}\sigma^*(t)$, may be written as:

$$\hat{\Gamma} = -\frac{1}{4}\sum_{q,p} J^q(\omega_q^p)\hat{\mathscr{D}}^\beta(\omega_p^q)[S_p^{-q}, S_p^q] \tag{61}$$

Here $\hat{\mathscr{D}}^\beta$ is the *thermalized double commutator superoprator*, that composes the superoperator from the the two operators $A, B$ in the way:

$$\hat{\mathscr{D}}^\beta(\omega)[A, B] = [A, f_\beta(\omega)B \bullet - \bullet Bf_\beta(-\omega)], \tag{62}$$

where $f_\beta(\omega) = \frac{2}{1+e^{\beta\omega}}$, and the dot is the place for operator on which superoperator is applied. It is easy to see that when $T \to \infty$, then $f_\beta(\omega) \to 1$ and the $\hat{\mathscr{D}}^\beta(\omega)$ becomes a double commutator superoperator, and the equation 61 becomes a standard sum of double commutator superoperators. This equation is completely equivalent to Eqs 43 and 48. Nevertheless, the form Lindblad dissipator is still easily recognizable, as one may substitute 56 to 43 and get:

$$\dot{\sigma}_{\rm S}^*(t) = \sum_{p,q} f_\beta(\omega_p^q)J^{q,-q}(\omega_p^q)\hat{\mathscr{L}}\mathscr{D}[S_p^{-q}, S_p^q]\sigma_{\rm S}^*(t).$$

Equation 59 partially decouples the statistical and the dynamical properties of the heat reservoir. Statistical properties, unrelated to the dynamics, which are functions of the temperature, are contained in the temperature factors, whereas the information about quantum-mechanical bath dynamics is contained in the Fourier transform of the symmetrized (here $\frac{1}{2}(J_R^{q,-q} + J_L^{-q,q})$) correlation functions. However, the latter still implicitly depend on the temperature through the trace over the bath degrees of freedom.

**The case of real correlation functions**

The above form of the relaxation master equation (Eqs. 59-61) is suitable for semi-classical approximations of relaxation where classical correlation functions can be used instead of quantum ones that are in general impossible to calculate or compute. It

is often the case that the classical correlation functions, calculated from classical models of dynamics, such as diffusion, jumps, ... are real functions of time. The condition of Eq. 51 then implies that the spectral density function $J^q(\omega)$ is even: $J^q(-\omega) = J^q(\omega)$.

## 5 Simplifications in the high temperature approximation

When the largest eigenvalue of the operators $S_p^q$ is such that $max(\beta\omega_p^q) \ll 1$, Eq. 59 takes the simpler form:

$$\dot{\sigma}_s^*(t) = \frac{1}{4}\sum_q -J^q(\omega_p^q)[S_p^{-q},[S_p^q,\sigma(t)]] + \frac{\beta\omega_p^q}{2}J^q(\omega_p^q)[S_p^{-q},\{S_p^q,\sigma(t)\}] \tag{63}$$

where $\{.,.\}$ denotes the anticommutator.

## The "low order" approximation

The assumption that the density operator is always close to the fully disordered state: $||\sigma - \frac{1}{A}|| \ll 1$, where $A$ is the dimension of the density operator, was made by Redfield (Redfield (1957)). Limiting the expansion to zeroth order, Eq. 63 becomes (Hubbard (1961)):

$$\dot{\sigma}_s^*(t) = \frac{1}{4}\sum_q -J^q(\omega_p^q)[S_p^{-q},[S_p^q,\sigma(t)]] + \frac{\beta\omega_p^q}{A}J^q(\omega_p^q)[S_p^{-q},S_p^q] \tag{64}$$

Moreover, using the property, Eq. 30, and the Taylor expansion of the exponential, it is straighforward to show that:

$$e^{-\beta H_s}S_p^q e^{\beta H_s} = e^{-\beta\omega_p^q}S_p^q \tag{65}$$

Therefore, when the density operator is in thermal equilibrium determined by the hamiltonian $H_s$, $\sigma^{eq} = \text{tr}_B(\exp^{-\beta H_s})^{-1}\exp^{-\beta H_s}$, one can show that $\mathcal{R}\sigma^{eq} = 0$, where $\mathcal{R}$ is defined by Eq. 59. Then, discarding terms that are second order or higher in $max(\beta\omega_p^q) \ll 1$, in Eq. 63, one gets the semi-classical formulation of the Redfield equation (Abragam (1961)):

$$\frac{d}{dt}(\sigma_s^*(t) - \sigma^{eq}) = -\frac{1}{2}\sum_{q,p}J^q(\omega_p^q)[S_p^{-q},[S_p^q,\sigma(t)-\sigma^{eq}]] \tag{66}$$

The evolution of the expected value of an operator is given by the alternative master equation:

$$\frac{d}{dt}\langle\mathcal{O}\rangle = \frac{1}{4}\sum_q -J^q(\omega_p^q)\langle[S_p^q,[S_p^{-q},\mathcal{O}]]\rangle + \beta\omega_p^q\sum_q J^q(\omega_p^q)\langle\{S_p^q,[\mathcal{O},S_p^{-q}]\}\rangle \tag{67}$$

In this expression, the second term on the right hand side contains the "thermal" contributions to relaxation, and can be selectively neglected for terms that are higher than first order in $\beta\omega_p^q$. That is, each term in the development such that:

$$\text{tr}\left(\{S_p^q,[\mathcal{O},S_p^{-q}]\}\sigma(t)\right) \ll \min(\text{tr}\left([S_p^q,[S_p^{-q},\mathcal{O}]]\sigma(t)\right)) \tag{68}$$

at all times can be discarded. Eq. 68 is in principle a less stringent condition and may provide criteria for the quasi-, pseudo-classical approximation - a test that can be verified a posteriori. It may thus provide a way to select which parts of the density operator can be discarded (neglected) and which must be retained in order to get an approximate analytical solution.

### The simple case of a two-spin system

 **"Double commutator" versus "thermal" contributions**

The differences of contributions between the first ("double commutator") and the second ("thermal") series of terms in equation 67 are illustrated in Figures 1 and 2 in the case of a pair of like spins $\frac{1}{2}$ subject to relaxation caused by a mutual dipolar (dipole-dipole - DD) interaction and the presence of a randomly fluctuating field (*ran*). Simulations were performed assuming Lorentzian spectral density functions:

$$J^{cl}_{DD,ran}(\omega) = \frac{2\tau_{C,ran}}{1+\omega^2\tau^2_{C,ran}} \tag{69}$$

with correlation times $\tau_{ran} = 60$ ps and $\tau_C = 8$ ps, for the random field and dipolar interactions, respectively. The dipolar coupling constant $b_{12} = -(\mu_0/4\pi)\gamma_I^2\hbar r_{12}^{-3}$ refers to the dipole-dipole coupling constant, $\gamma_I$ the gyromagnetic ratio and $r_{12}$ the internuclear distance. In these simulations, $b_{12} = 35 \cdot 10^3$ Hz. These values were chosen to give $T_1 \approx 2$ s and $T_S \approx 20$ s.

Contributions from both relaxation mechanisms to the expected values in Eq. 67 were computed, for the spins prepared either in a singlet state, or inverted from thermal equilibrium. The individual terms entering the first and second sums in the rhs of Equation 67 are depicted for the case of the magnetization $\mathcal{O} = I_z + S_z$ (Fig. 1) and singlet state $\mathcal{O} = \frac{1}{4}\mathbf{1} - \mathbf{I}\cdot\mathbf{S}$ (Fig. 2) operators. The time evolutions of all the contributions to the rate of change of the expected value of the operator $\mathcal{O}(t)$ are depicted. When the spins are initially prepared in the state $-(I_z + S_z)$, the thermal contribution (blue curve) to the rate of change of the magnetization has no effect and the only contribution to $\frac{d\langle\mathcal{O}\rangle(t)}{dt}$ comes from the double commutator (red curve). This is the case for both relaxation, dipolar and random field fluctuations, mechanisms (Figs 1(a) and (b)). Moreover, the values chosen for the simulation imply that the dipolar contribution (of the double commutator in this case) to the total relaxation rate $< \dot{\mathcal{O}} > (t)$ is much larger than than the one of the random field.

The situation is strikingly different when the spins are initially prepared in a singlet order. Here, the thermal correction (blue) is negligible with respect to the double commutator (red) contribution to the rate of change $< \dot{\mathcal{O}} > (t)$ only for the dipole-dipole mechanism (Fig. 1(c)). In contrast, for terms originating from random field relaxation, both thermal and dipolar terms are of comparable orders of magnitude (Fig. 1(c)). These are the terms that cannot be neglected in an approximate solution of Eq. 67. Fig. 1(c) also shows that the dipolar contribution to $< \dot{\mathcal{O}} > (t)$ increases with time, which is consistent with the progressive depletion of the singlet order (immune to dipolar relaxation). And for random relaxation, which mainly affects the singlet order in this example, Fig. 1(d) illustrates the fact that the weight of the thermal contribution decays with time, with the concomitant increase of the double commutator term, which is also due to the progressive depletion of the singlet order.

The situation depicted in Fig. 2 is different, and shows the the rate of change of the expected value $< \mathcal{O} > (t)$ where $\mathcal{O} = \frac{1}{4}(\mathbf{I}\cdot\mathbf{S})$, for the same initial state conditions as above. In this case, the dipole-dipole simply does not contribute to $< \dot{\mathcal{O}} > (t)$, as expected from symmetry considerations. This is of course the case whatever the initial state (inverted magnetization (Fig.2(a)) or singlet order (Fig.2 (c))). This illustrates the known fact that singlet state is immune to dipolar relaxation for symmetry reasons.

Alternatively, when the spins are prepared in the $-(I_z + S_z)$ state, both thermal and double commutators contribute, albeit to a negligible amount, showing that the spins evolve mostly towards magnetization (compare the scales with Fig. 1(b)), and that only a negligible part is transferred to singlet order.

Interestingly, Fig. 2(d) shows that there is no thermal contribution (blue) to the rate $< \dot{\mathcal{O}} > (t)$, and that, starting from a singlet order, its evolution can be predicted by discarding the thermal thermal terms of Eq. 67, and therefore retaining the simple, double commutator, expression for the relaxation master equation.

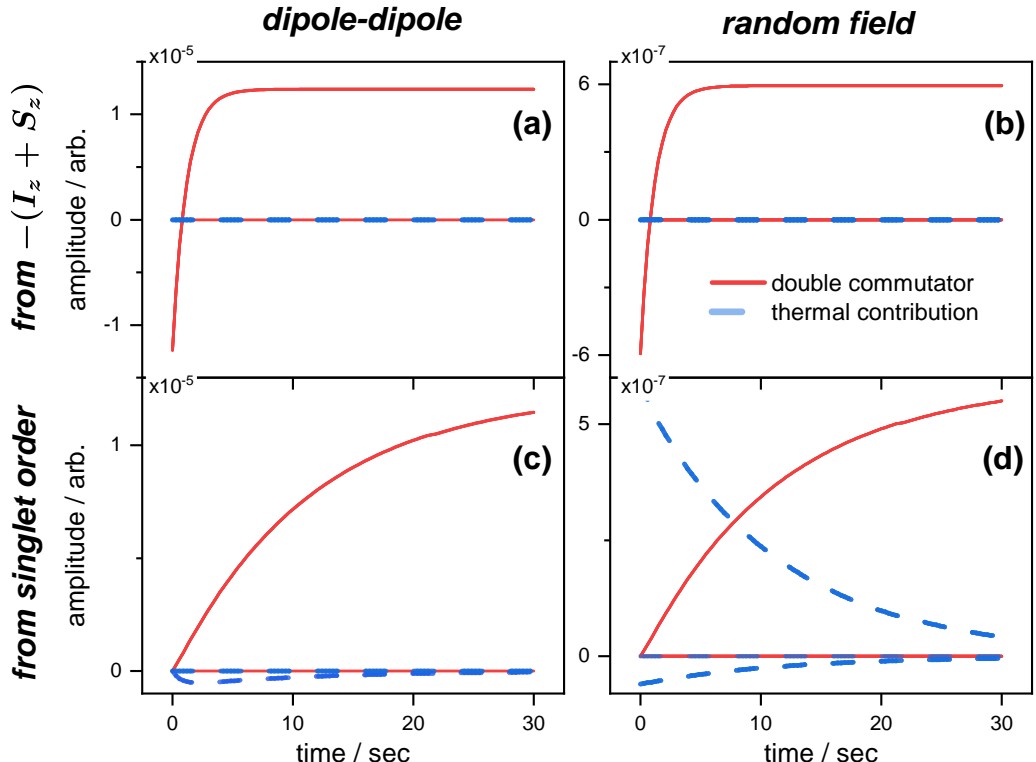

**Figure 1.** Expected values of magnetization spin operator $\mathcal{O} = I_z + S_z$: contributions of the "double commutator" (red curves) and "thermal" (blue curves) parts of the Redfield relaxation operator to the magnetization from the different operators $S_p^q$ (see Table C1). (a) and (b) correspond respectively to the dipolar and random field relaxation of spins inverted from a Boltzmann equilibrium; (c) and (d) correspond respectively to the dipolar and random field relaxation of spins initially prepared in a singlet state.

**Singlet-triplet conversion**

The recent achievement of the Lindblad approach was the description of the magnetization relaxation of a two-spin system prepared in a singlet state (Bengs and Levitt (2020)). In that paper, detailed balance was enforced through the Schofield

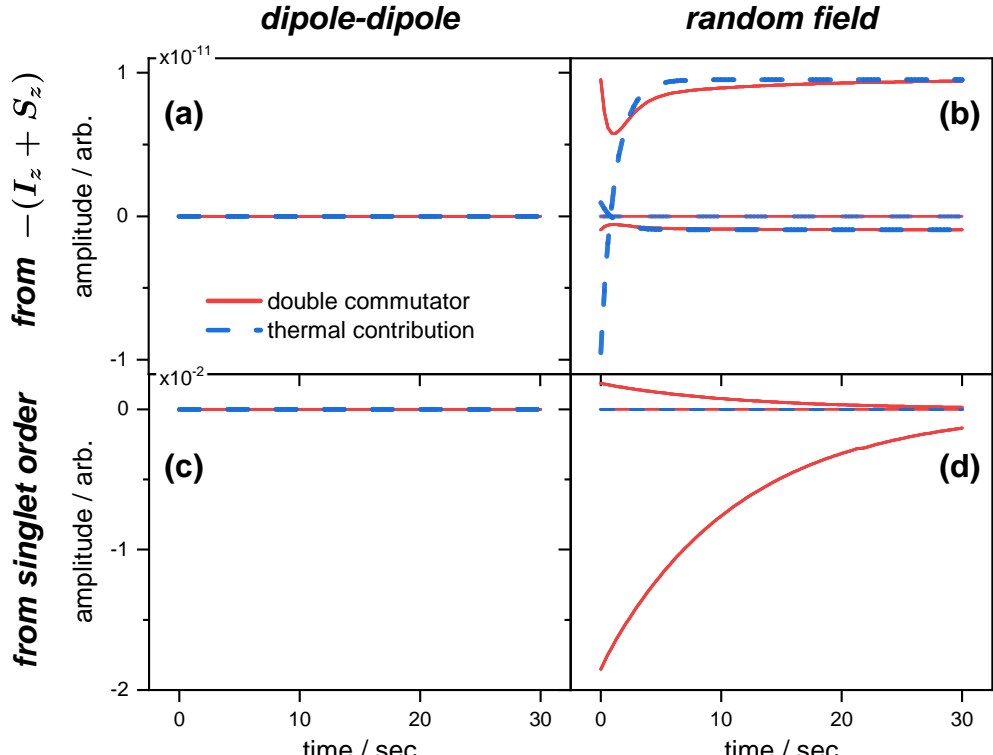

**Figure 2.** Same as Fig. 1 for expected value of the singlet operator $\mathcal{O} = \frac{1}{4}\mathbf{1} - \mathbf{I} \cdot \mathbf{S}$.

procedure (Schofield (1960)), whereby spectral density functions are built from classical ones through the transformation:

$$J(\omega)_{L,R} \rightarrow J^{cl}(\omega)e^{-\frac{\beta\omega}{2}}, \tag{70}$$

where $J^{cl}(\omega)$ refers to the classical spectral density function. Eq. 70 is one among several that have been proposed to make classical spectral density functions asymmetric so as to obey the detailed balance condition (White et al. (1988); Egorov and Skinner (1998); Egorov et al. (1999); Ramirez and López-Ciudad (2004); Frommhold (1993)). In Ref. (Bengs and Levitt (2020)), $J(\omega)$ was assumed to be any kind of spectral density function obtained through classical models, such as diffusion jumps, etc. The distinction between left and right spectral densities that appears in the course of the conventional perturbative derivation of the master equation, was not made there. Moreover, the detailed balance condition appears as an additional requirement, as this condition is not implied by Linblad's approach that merely provides the general mathematical structure of the evolution equation obeyed by the density operator that complies with the requirements of quantum mechanics, in the presence of a Markovian dissipative process (Lindblad (1976); Alicki and Lendi (2007)).

In the following, we derive the evolution of the magnetization of a two-spin system using the singlet-triplet population basis and compare results obtained by both approaches. As above (and in Ref. (Bengs and Levitt (2020))) the relaxation superoperator $\hat{\Gamma}$ is the sum of contributions from mutual dipole-dipole relaxation ($\hat{\Gamma}_{DD}$) and the interaction with a partially correlated random field ($\hat{\Gamma}_{ran}$):

$$\hat{\Gamma} = \hat{\Gamma}_{DD} + \hat{\Gamma}_{ran}, \tag{71}$$

The irreducible tensor operator $T_{\lambda\mu}$ representation is better suited to derive analytical solutions for the problem at hand, where $\hat{\Gamma}$ is expressed, according to Eq. 62, as:

$$\hat{\Gamma}_{DD} = \frac{6}{5}b_{12}^2 \sum_{\mu=-2}^{\mu=2} J_{DD}^{cl}(\mu\omega^0)\hat{\mathcal{D}}^\beta(\mu\omega^0)[T_{2\mu}^{(12)\dagger}, T_{2\mu}^{(12)}],$$

$$\hat{\Gamma}_{ran} = \sum_{i,j=1}^{2} \kappa_{ij}\omega_{rms}^{(i)}\omega_{rms}^{(j)} \sum_{\mu=-1}^{\mu=1} J_{ran}^{cl}(\mu\omega^0)\hat{\mathcal{D}}^\beta(\mu\omega^0)[T_{1\mu}^{(i)\dagger}, T_{1\mu}^{(j)}]. \tag{72}$$

The $T_{\lambda\mu}$ are eigenoperators of the main Zeeman Hamiltonian and their expressions are shown in Appendix C. $\omega_{rms}^{(i)}$ is the root mean square fluctuation of the random field acting on spin $I_i$, and in the isotropic case considered here, is identical for both nuclei, so that $\omega_{rms}^{(1)} = \omega_{rms}^{(2)}$. The coefficient $-1 \leq \kappa_{12} \leq 1$ describes the degree of correlation of random field fluctuations on the 1 and 2 nuclei. By definition, $\kappa_{11} = \kappa_{22} = 1$. In order to simplify the notations, we will henceforth drop the subscript ($\kappa_{12} \to \kappa$).

In the extreme narrowing regime where $\omega\tau_{c,ran} \ll 1$, the spectral densities become frequency independent and $J(\omega) \approx 2\tau$. The dipole-dipole and random field contributions to the longitudinal relaxation rate constant $R_1 = R_1^{DD} + R_1^{ran}$ are given by, according to Eq. 62:

$$R_1^{DD} = -\frac{(I_z|\hat{\Gamma}_{DD}|I_z)}{(I_z|I_z)} = \frac{3}{20}b_{12}^2\tau_c\left(4f_\beta(2\omega_0) + 4f_\beta(-2\omega_0) + f_\beta(\omega_0) + f_\beta(-\omega_0)\right) = \frac{3}{2}b_{12}^2\tau_c,$$

$$R_1^{ran} = -\frac{(I_z|\hat{\Gamma}_{ran}|I_z)}{(I_z|I_z)} = \omega_{rms}^2\tau_{ran}\left(f_\beta(\omega_0) + f_\beta(-\omega_0)\right) = 2\omega_{rms}^2\tau_{ran} \tag{73}$$

where $f_\beta(\omega) = \frac{2}{1+e^{\beta\omega}}$. It is interesting to note that in this model the relaxation rates do not depend on the temperature, in contrast to Ref. (Bengs and Levitt (2020)). This is not due to any approximation, rather, it arises from the fact that $f_\beta(\omega) + f_\beta(-\omega) = 2$, which is explicit in equation 73. Similarly, the singlet order relaxation rate is given by:

$$R_S = R_S^{ran} = -\frac{(\mathbf{I}_1\mathbf{I}_2|\hat{\Gamma}_{ran}|\mathbf{I}_1\mathbf{I}_2)}{(\mathbf{I}_1\mathbf{I}_2|\mathbf{I}_1\mathbf{I}_2)} = 4\omega_{rms}^2\tau_{ran}(1-\kappa) \tag{74}$$

, which does not depend on the temperature. For sake of comparison with the results of (Bengs and Levitt (2020)), we use the singlet-triplet population basis, where the singlet and triplet states are defined as:

$$|S_0\rangle = (|\alpha_1\beta_2\rangle - |\beta_1\alpha_2\rangle)/\sqrt{2}, \tag{75}$$

$$|T_{+1}\rangle = |\alpha_1\alpha_2\rangle,$$

$$|T_0\rangle = (|\alpha_1\beta_2\rangle + |\beta_1\alpha_2\rangle)/\sqrt{2},$$

$$|T_{-1}\rangle = |\beta_1\beta_2\rangle,$$

where $|\alpha\rangle$ and $|\beta\rangle$ denote the Zeeman spin states of an isolated spin-1/2 nucleus with $z$-projection of +1/2 and –1/2. In this representation, the population block of relaxation superoperator 72 is given by:

$$[\hat{\Gamma}]_{4\times 4} =$$

| | $|S_0\rangle\langle S_0|$ | $|T_{+1}\rangle\langle T_{+1}|$ | $|T_0\rangle\langle T_0|$ | $|T_{+1}\rangle\langle T_{+1}|$ |
|---|---|---|---|---|
| | $-\Sigma_1$ | $\frac{1}{2}(1-\kappa)R_1^{ran}f_\beta(-\omega_0)$ | $\frac{1}{2}(1-\kappa)R_1^{ran}$ | $\frac{1}{2}(1-\kappa)R_1^{ran}f_\beta(\omega_0)$ |
| | $\frac{1}{2}(1-\kappa)R_1^{ran}f_\beta(\omega_0)$ | $-\Sigma_2$ | $\frac{1}{5}R_1^{DD}f_\beta(\omega_0)+$ $\frac{1}{2}R_1^{ran}(1+\kappa)f_\beta(\omega_0)$ | $\frac{2}{5}R_1^{DD}f_\beta(2\omega_0)$ |
| | $\frac{1}{2}(1-\kappa)R_1^{ran}$ | $\frac{1}{5}R_1^{DD}f_\beta(-\omega_0)+$ $\frac{1}{2}R_1^{ran}(1+\kappa)f_\beta(-\omega_0)$ | $-\Sigma_3$ | $\frac{1}{5}R_1^{DD}f_\beta(\omega_0)+$ $\frac{1}{2}R_1^{ran}(1+\kappa)f_\beta(\omega_0)$ |
| | $\frac{1}{2}(1-\kappa)R_1^{ran}f_\beta(-\omega_0)$ | $\frac{2}{5}R_1^{DD}f_\beta(-2\omega_0)$ | $\frac{1}{5}R_1^{DD}f_\beta(-\omega_0)+$ $\frac{1}{2}R_1^{ran}(1+\kappa)f_\beta(-\omega_0)$ | $-\Sigma_4$ |

(76)

where $\Sigma_i$ denotes the sum of the terms alongside respective column. This matrix is very similar to one introduced in (Bengs and Levitt (2020)), but with the substitution of term $\theta(\omega) = \exp(-\beta\omega/2)$ by $f_\beta(\omega)$.

In the high-temperature limit, both terms become approximately equal, $\theta(\omega) \approx f_\beta(\omega) \approx 1 - \frac{\beta\omega}{2}$ and the difference between $\theta(\omega)$ and $f_\beta(\omega)$ becomes significant only in case of extremely low temperatures or very high frequencies. As could be expected in this limit, the time evolution of the z magnetization is given by the same biexponential behavior as in Ref. (Bengs and Levitt (2020)):

$$\langle I_z(t)\rangle / \langle I_z^{eq}\rangle \approx 1 + A_1 e^{-R_1 t} + A_S e^{-R_S t} \tag{77}$$

with the same coefficients $A_1 = \dfrac{R_S}{2(R_1 - R_s)}$ and $A_S = \dfrac{-2R_1 + R_S}{2(R_1 - R_s)}$. The fact that $A_1$ and $A_S$ are the ones found in (Bengs and Levitt (2020)) is expected, because in this limit $R_1$ and $R_s$ do not depend on the temperature factor. In the usual conditions of high but not infinite temperature, it is found that the next nonzero terms in the expansions of $A_1(\beta)$ and $A_S(\beta)$ are of degree $\beta^2$, and therefore do not contribute in the regime where $\omega\beta \ll 1$. These expressions can be found in Appendix D.

The foregoing discussion has shown that in the high temperature approximation, the exact "thermalization" procedure of the spectral density function is irrelevant, as all models are equivalent in these conditions. Indeed, in a field of 23.5 T (1000 MHz resonance proton frequency) the temperature at which $\hbar\omega_H \approx kT$, where both approaches may lead to significant differences, is $T \approx 50$ mK. These are unrealistic experimental conditions. Alternatively, the master equation of Eqs. (59-62) may well be of use in the context of DNP to describe the electron spin-lattice relaxation outside of the high-temperature limit, through direct spin-phonon coupling at temperatures below 4 K and at high fields, where this process is predominant.

## 6   Conclusion: a remark on the "semiclassical" theory

In the semiclassical viewpoint (as in Ref. (Abragam (1961)), for instance), the effect of the bath is taken into account through a stochastic spin hamiltonian, the spatial part of which is a function of the lattice variables and is a random function of time. It

is usually understood that this approach does not comply with the Boltzmann equilibrium of the bath. Besides, the stationary state reached by the spin density operator is left undetermined by the master equation so that it must be enforced by the supplementary *ad hoc* assumption that the spins return to the Boltzmann distribution of the spin populations. A recent analysis by Bengs and Levitt (Bengs and Levitt (2020)) showed that the usual semi-classical master equation was not able to predict the correct magnetization evolution of a two-spin system prepared in a singlet state. Thus, the usual semiclassical inhomogeneous master equation provides erroneous predictions in this case. The latter is obtained when the thermal corrections to the double commutator part of the relaxation operator are retained to first order in the largest eigenvalue $\frac{\omega_p^q}{kT}$, and the relaxation operator reduces to a double commutator ("low order" case). However, the equilibrium density operator is not a stationary solution in this case and therefore a correction term is added to the master equation, leading to the same result as the usual semiclassical master equation (Hubbard (1961)). And so, when the low order assumption is not verified, as in the case of a spin system prepared in the singlet state, this description becomes inconsistent.

As shown above, the non commutation of the bath operators have critical consequences, leading to the lattice temperature dependent terms in the master equation, and only when the bath operators $[B^{-q}(t), B^q(0)] = 0$ one recovers the double commutator expression, with the additional property $J_{\text{L}}(\omega) = J_{\text{R}}(\omega)$, so that the Kubo relation imposes an infinite lattice temperature. This illustrates how the finite temperature of the lattice is conveyed to the spins through non commutation of the bath operators of the coupling hamiltonian.

The conventional semiclassical approach, where spin-bath interactions are represented by random spin hamiltonians, has two simultaneous consequences: the structure of the relaxation operator is affected in such a way that the master equation takes the form of a double commutator; and since $J_{\text{L}}(\omega) = J_{\text{R}}(\omega)$, the system cannot evolve to a thermodynamic equilibrium associated with a finite temperature. In this case, detailed balance property is conserved but only in the special case of infinite lattice temperature. In fact, since detailed balance is statistical by nature, it is *per se* compatible with a semiclassical approach. If, on the other hand, detailed balance is taken into account in the semi-classical theory of NMR relaxation, so that $J(-\omega) = e^{-\hbar\omega/kT} J(\omega)$ and the general relations Eq. 49 or 51 obeyed by correlation functions are retained, it is easy to show from the symmetry properties of the spectral density function that the semiclassical Redfield equation (Abragam (1961)):

$$\dot{\sigma}_{\text{S}}^*(t) = -\frac{1}{2} \sum_{q,p} J(\omega_p^q)[S_p^{-q}, [S_p^q, \sigma(t)]] \tag{78}$$

is obeyed, with this definition of $J(\omega)$. However, the expected equilibrium density operator is not a stationary state of Eq. 78 in this case, which illustrates the (also known) fact that this condition alone is insufficient to completely determine the transition probabilities of the bath in the absence of a dynamical model for the latter. On the other hand, it is possible to describe the dynamics of a classical system where microscopic irreversibility, i.e., detailed balance, is ensured. This is straightforward from the definition of the correlation function of a phase variable in classical mechanics, $\langle \mathbf{A}(t)\mathbf{B}^* \rangle = \int dq dp \rho^e B^* e^{i\mathbf{L}t} A$, where $\mathbf{L}$ is the classical Liouvillian acting on the phase space (Evans and Moriss (2008)). In addition, general procedures have been used that provide Fokker-Planck or master equations for diffusion that obey the detailed balance condition, yielding classical spectral density functions that comply with the Boltzmann equilibrium distribution and the classical laws of motion of the bath (see for instance (VanKampen (1981); Risken (1972); Wassam et al. (1980))), in particular in the context of magnetic resonance

(Stillman and Freed (1980)). As stated in several instances in this work, in the fully quantum approach, detailed balance is ensured by assuming that the bath is in a stationary state defined by a Boltzmann distribution of its energy states. Thus, the "irreducible" difference between the semi-classical and the fully quantized theory lies in the fact that the bath operators do not mutually commute, which prevents the expression in Eq. 27 to reduce to the double commutator. Both thermodynamic and quantum mechanical effects are thus entangled in the fully quantum mechanical treatment of relaxation.

*Financial support.* B.R. acknowledges a Vernadski scholarship from the French embassy in the Russian Federation. D.A. acknowledges a PRC (Projet de Recherche Collaborative) funding from the Centre National de la Recherche Scientifique. This research has been supported by the Russian Foundation for Basic Research (grant no. 20-53-15004)

*Author contributions.* DA designed the research, DA and BR performed theoretical derivations, simulations and wrote the paper.

*Competing interests.* The authors declare that they have no conflict of interest.

## Appendix A: Derivation of Eq. 19

$$\frac{d}{dt}\sigma(t) \;=\; \mathcal{L}_{\mathrm{S}}\sigma(t) + e^{\mathcal{L}_{\mathrm{S}}t}\int_0^t dt'\,\mathrm{tr}_{\mathrm{B}}\left\{\mathcal{L}_1^*(t)\mathcal{L}_1^*(t')\rho_{\mathrm{B}}^e\right\}\sigma^*(t) \tag{A1}$$

$$=\; \mathcal{L}_{\mathrm{S}}\sigma(t) + \int_0^t dt'\,\mathrm{tr}_{\mathrm{B}}\left\{e^{\mathcal{L}_{\mathrm{S}}t}\mathcal{L}_1^*(t)\mathcal{L}_1^*(t')\rho_{\mathrm{B}}^e\right\}e^{-tL_{\mathrm{S}}}\sigma(t) \tag{A2}$$

$$=\; \mathcal{L}_{\mathrm{S}}\sigma(t) + \int_0^t dt'\,\mathrm{tr}_{\mathrm{B}}\left\{e^{\mathcal{L}_{\mathrm{S}}t}e^{-(\mathcal{L}_{\mathrm{S}}+\mathcal{L}_{\mathrm{B}})t}\mathcal{L}_1 e^{(\mathcal{L}_{\mathrm{S}}+\mathcal{L}_{\mathrm{B}})t}e^{-(\mathcal{L}_{\mathrm{S}}+\mathcal{L}_{\mathrm{B}})t'}\mathcal{L}_1 e^{(\mathcal{L}_{\mathrm{S}}+\mathcal{L}_{\mathrm{B}})t'}e^{-tL_{\mathrm{S}}}\rho_{\mathrm{B}}^e\right\}\sigma(t) \tag{A3}$$

$$=\; \mathcal{L}_{\mathrm{S}}\sigma(t) + \int_0^t dt'\,\mathrm{tr}_{\mathrm{B}}\left\{e^{-\mathcal{L}_{\mathrm{B}}t}\mathcal{L}_1 e^{(\mathcal{L}_{\mathrm{S}}+\mathcal{L}_{\mathrm{B}})t}e^{-(\mathcal{L}_{\mathrm{S}}+\mathcal{L}_{\mathrm{B}})t'}\mathcal{L}_1 e^{(\mathcal{L}_{\mathrm{S}}+\mathcal{L}_{\mathrm{B}})t'}e^{-tL_{\mathrm{S}}}\rho_{\mathrm{B}}^e\right\}\sigma(t) \tag{A4}$$

$$=\; \mathcal{L}_{\mathrm{S}}\sigma(t) + \int_0^t dt'\,\mathrm{tr}_{\mathrm{B}}\left\{\mathcal{L}_1 e^{(\mathcal{L}_{\mathrm{S}}+\mathcal{L}_{\mathrm{B}})t}e^{-(\mathcal{L}_{\mathrm{S}}+\mathcal{L}_{\mathrm{B}})t'}\mathcal{L}_1 e^{(\mathcal{L}_{\mathrm{S}}+\mathcal{L}_{\mathrm{B}})t'}e^{-tL_{\mathrm{S}}}e^{-\mathcal{L}_{\mathrm{B}}t}\rho_{\mathrm{B}}^e\right\}\sigma(t) \tag{A5}$$

$$=\; \mathcal{L}_{\mathrm{S}}\sigma(t) + \int_0^t dt'\,\mathrm{tr}_{\mathrm{B}}\left\{\mathcal{L}_1 e^{-(\mathcal{L}_{\mathrm{S}}+\mathcal{L}_{\mathrm{B}})(t'-t)}\mathcal{L}_1 e^{(\mathcal{L}_{\mathrm{S}}+\mathcal{L}_{\mathrm{B}})(t'-t)}\rho_{\mathrm{B}}^e\right\}\sigma(t) \tag{A6}$$

In this derivation, the invariance of the trace to circular permutations has been used, as well as the fact that $[\mathcal{L}_{\mathrm{B}}, \rho_{\mathrm{B}}^e] = 0$, since the bath is in a stationary state.

## Appendix B: Evaluation of the terms of eq. 44

The correlation functions involved in Equation 44:

$$\frac{1}{2}J_{\mathrm{R}}^{q,-q}(\omega_p^q) \;=\; \int_0^\infty d\tau\,\langle B^q B^{-q}(\tau)\rangle e^{-i\omega_p^q\tau}$$

$$=\; \int_0^\infty d\tau\,\frac{1}{L}\sum_{f,f'}\langle f|B^q|f'\rangle\langle f'|B^{-q}|f\rangle e^{i(f'-f)\tau}e^{-\beta f}e^{-i\omega_p^q\tau} \;=\; \int_0^\infty d\tau\,\frac{1}{L}\sum_{f,f'}|\langle f|B^q|f'\rangle|^2 e^{i(f'-f)\tau}e^{-\beta f}e^{-i\omega_p^q\tau}$$

$$\approx\; \frac{1}{2}\frac{1}{L}\sum_{f,f'}|\langle f|B^q|f'\rangle|^2 e^{-\beta f}\int_{-\infty}^\infty e^{i(f'-f-\omega_p^q)\tau}\,d\tau \;=\; \frac{1}{2}\frac{1}{L}\sum_{f,f'}|\langle f|B^q|f'\rangle|^2 e^{-\beta f}\delta(f'-f-\omega_p^q)$$

$$=\; \frac{1}{2}\frac{1}{L}\sum_{f}|\langle f|B^q|f+\omega_p^q\rangle|^2 e^{-\beta f} \tag{B1}$$

since $B^{-q} = B^{q\dagger}$. Similarly, one has:

$$
\begin{aligned}
\frac{1}{2} J_{\mathrm{L}}^{-q,q}(\omega_p^q) &= \int_0^\infty d\tau \langle B^{-q}(\tau) B^q \rangle e^{-i\omega^q \tau} = \frac{1}{L} \sum_{f,f'} \langle f|B^{-q}|f'\rangle \langle f'|B^q|f\rangle e^{-\beta f} \int_0^\infty d\tau e^{i(f-f')\tau} e^{-i\omega^q \tau} \\
&= \frac{1}{2}\frac{1}{L} \sum_{f,f'} |\langle f|B^{-q}|f'\rangle|^2 e^{-\beta f} \int_{-\infty}^\infty e^{i(f-f'-\omega^q)\tau} = \frac{1}{2}\frac{1}{L} \sum_{f,f'} |\langle f|B^{-q}|f'\rangle|^2 e^{-\beta f} \delta(f - f' - \omega^q)
\end{aligned}
$$

Noting that $|\langle f|B^{-q}|f'\rangle| = |B_{ff'}^{-q}| = |B_{f'f}^{q\dagger}| = |B_{f'f}^{q*}| = |B_{f'f}^q|$, and exchanging indices $f \leftrightarrow f'$, one gets, from Eq. B2:

$$
\begin{aligned}
\frac{1}{L} \sum_{f,f'} |\langle f|B^{-q}|f'\rangle|^2 e^{-\beta f} \delta(f - f' - \omega_p^q) &= \frac{1}{L} \sum_{f,f'} |\langle f'|B^q|f\rangle|^2 e^{-\beta f} \delta(f - f' - \omega_p^q) \\
&= \frac{1}{L} \sum_{f,f'} |\langle f|B^q|f'\rangle|^2 e^{-\beta f'} \delta(f' - f - \omega_p^q) = \frac{1}{L} \sum_f |\langle f|B^q|f+\omega_p^q\rangle|^2 e^{-\beta(f+\omega_p^q)} \\
&= e^{-\beta\omega_p^q} \frac{1}{L} \sum_f |\langle f|B^q|f+\omega_p^q\rangle|^2 e^{-\beta f} = e^{-\beta\omega^q} J^{\mathrm{R}}(\omega_p^q)
\end{aligned} \tag{B2}
$$

Besides, it immediately follows from the definitions of $J_{\mathrm{R}}^q(\omega)$ and $J_{\mathrm{L}}^q(\omega)$ that:

$$
J_{\mathrm{R}}^q(-\omega) = J_{\mathrm{R}}^{-q}(\omega) \tag{B3}
$$

## Appendix C: Eigenoperators for a homonuclear coupled spin 1/2 pair

The case of homonuclear spin pair the main hamiltonian is defined as:

$$
H_Z = \omega^0 (I_{1z} + I_{2z}), \tag{C1}
$$

where $\omega^0 = -\gamma B$, and $\gamma$ is the magnetogyric ratio and $B$ is the field strength. The eigenoperators for this hamiltonain are summarized in the table C1. The eigenoperators are denoted by $T_{\lambda\mu}^{(ij)}$. The superscript $(ij)$ indicates angular momentum coupling of spins $I_i$ and $I_j$ resulting in a spherical tensor operator of total angular momentum $\lambda$ and $z-$angular momentum $\mu$

**Table C1.** Eigenoperators of hamiltonian for a homonuclear coupled spin-1/2 pair

| $\mu \backslash \lambda$ | 2 | 1 |
|---|---|---|
| $\pm 2$ | $\frac{1}{2} I_1^\pm I_2^\pm$ | $-$ |
| $\pm 1$ | $\mp \left( I_1^\pm I_2^z + I_{1z} I_2^\pm \right)$ | $\mp \frac{1}{\sqrt{2}} I_j^\pm$ |
| $0$ | $-\frac{1}{2\sqrt{6}} \left( I_1^+ I_2^- + I_1^- I_2^+ - 4 I_{1z} I_{2z} \right)$ | $I_{jz}$ |

## Appendix D: Expansion coefficient for the magnetization evolution solution

The coefficient from the equation 77 are in fact temperature dependent:

$$A_1(\beta) = A_1 + C_1\beta^2 + \mathcal{O}(\beta^4)$$
$$A_S(\beta) = A_S + C_S\beta^2 + \mathcal{O}(\beta^4)$$

(D1)

where the coefficients $A_1$ and $A_S$ was shown before in the main text, and the coefficients $C_1$ and $C_S$ is defined as:

$$C_1 = \omega_0^2 \frac{R_S\left(600R_1^3 - 20R_1^2(4416R_1^{dd} + 35R_S) - 5R_s(17280(R_1^{dd})^2 + 2958R_1^{dd}R_s + 5R_S^2) + 2R_1(43488(R_1^{dd})^2 + 51120R_1^{dd}R_S + 125R_S^2)\right)}{5760R_1(150R_1 - 149R_1^{dd} - 25R_S)(R_1 - R_S)^2}$$

(D2)

$$C_S = -C_1$$

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
