# Peer review of "Spin relaxation: under the sun, anything new?\"

_Magnetic Resonance, 2021_

## Referee Comment (RC1)

**RESPONSE TO "SPIN RELAXATION: UNDER THE SUN, ANYTHING NEW?" BY B.A. RODIN AND D. ABERGEL.**

The work by Rodin and Abergel revisits some of the early developments of spin relaxation theory in NMR and discusses their relation to the Lindblad equation, which is commonly employed in quantum optics and the mathematical physics literature. A detailed discussion on classical and quantum mechanical correlation functions/spectral densities with respect to detailed balance is also included.

The paper seems to be in response to a recent publication "A master equation for spin systems far from equilibrium" (doi.org/10.1016/j.jmr.2019.106645) authored by Malcolm Levitt and myself. However, several references and conclusions regarding our work are incorrect or misstated.

In their abstract Rodin and Abergel claim that our work has "recently questioned" the "underlying theory" worked out by the founding fathers of spin relaxation theory. This is incorrect, we clearly state at the beginning of the article that our work explores the disadvantage and advantages of several thermalization techniques (IME, Jeener, Levitt-di-Bari) in the context of highly polarised spin systems (see section 4). In particular we show that the inhomogeneous master equation (IME) may lead to non-physical results for such systems. At no point do we cast doubt upon the underlying theory developed by Bloch, Wangness, Redfield, etc. On the contrary, we give full credit to the "founding fathers" already being "fully aware that semiclassical relaxation theory [...] is of limited validity" for highly polarised systems. If anything we cast doubt upon the current use of the IME for highly polarised spin systems within the NMR community.

Within this context section 2.2 by Rodin and Abergel seems to stipulate that we imply that it is impossible to derive the Lindblad equation from the quantized theory of Bloch and Redfield. However, we simply presented the Lindblad method as a convenient approach to extend the semiclassical relaxation theory to systems outside the weak-order limit. We rather hoped that it would be clear that the Lindblad equation follows from the quantized BWR theory given that the starting point of our derivation in section 5 coincides with the starting point of BWR. However, it is true that we could have stated this fact in more detail as has recently been done by Tom Barbara (https://doi.org/10.5194/mr-2-689-2021). The authors conclude section 2.2 with "the Lindblad equation [...] is thus derived from the usual

quantized theory of relaxation." I am not sure if the authors are trying to claim novelty for this result (it also appears to be core statement of their abstract), if so they should consult the books by Breuer (DOI:10.1093/acprof:oso/9780199213900.001.0001), Schaller (10.1007/978-3-319-03877-3), and the recent article by Tom Barbara.

Unfortunately some mischaracterisation of our analysis of singlet-triplet relaxation in highly polarised spin systems may be found within the conclusion. The authors state: "The author's interpretation was that "thermalization" was not properly taken into account by the inhomogeneous semiclassical version of the Redfield equation, which was ascribed to the fact that the classical approach does not comply with the detailed balance condition." No such statements are made. The deviation between these results was not ascribed to detailed balance, but to the fact that the IME generates the unphysical situation in which singlet order may directly flow into Zeeman magnetisation (clearly illustrated in figure 1). The provided discussion by the authors on this subject is also somewhat confusing. If I understand correctly the authors strongly emphasise the validity of detailed balance for the bath (in example the equilibrium distribution of the liquid follows Boltzmann). But this does not matter very much if the *spin transitions* do not obey detailed balance, which is clearly what our work refers to when talking about detailed balance. Consider for example equation (77) with a spectral density $J^\theta(\omega) = \theta(\omega)J(\omega)$, where $\theta(\omega)$ enforces detailed balance of the *spectral densities*. Consider now an ensemble of spin-1/2 particles experiencing random field relaxation in solution. According to equation (77) the relaxation superoperator is given by

$$\hat{\Gamma} = -\frac{1}{2}\xi_{\mathrm{erf}} \sum_{m=-1}^{+1} J(|m|\omega)\theta(m\omega)\hat{T}_{1m}^\dagger \hat{T}_{1m}.$$ (1)

It is straightforward to verify the following matrix elements

$$W_{\alpha\leftarrow\beta} = (P_\alpha|\hat{\Gamma}|P_\beta) = \frac{\xi_{\mathrm{erf}}}{4}J(\omega)(\theta(+\omega) + \theta(-\omega))$$
$$W_{\beta\leftarrow\alpha} = (P_\beta|\hat{\Gamma}|P_\alpha) = \frac{\xi_{\mathrm{erf}}}{4}J(\omega)(\theta(+\omega) + \theta(-\omega)),$$ (2)

where $P_\alpha$ and $P_\beta$ are the $\alpha$ and $\beta$ population operators, respectively. The two transition rates per unit time $W_{\alpha\leftarrow\beta}$ and $W_{\beta\leftarrow\alpha}$ are *equal*. So even if the detailed balance condition holds for the bath it *does not* hold for the spin-state transitions when equation (77) is used. In other words it is correct to say that the equilibrium density operator for the spin system is not in the nullspace of $\hat{\Gamma}$ because it does not obey detailed balance for spin transitions. This has been discussed extensively in appendix A.2 of our work.

Besides the very unfortunate misinterpretation of our work, the provided discussion regarding classical and quantum mechanical correlation functions is interesting and is used to introduce a "thermalized double-commutator" in section 4. The thermalized double-commutator renders the relaxation superoperator Redfield like and it is shown that this approach is fully equivalent to the Lindblad method. However advantages and disadvantages, if there are any, are not discussed. For example, the thermalized double-commutator does not represent a double-commutator in a strict mathematical sense and might be awkward to incorporate into existing simulation software. Additionally, representing the relaxation superoperator in Lindblad form does not only provide a simple check on positivity and Markov compatibility, but is also in agreement with the majority of the literature outside NMR.

Apart from that there also some technical issues.

- The trace operator Tr should not be italicized.

- The interaction frame transformation above equation (4) should read as $\exp(-\mathcal{L}_0 t)$.

- The authors switch to a time difference variable $\tau = t - t'$ in equation (6), but this is not consistent with equations (7)-(19).

- Something appears to be wrong with equation (19), take for example $\mathcal{L}_1 = A \otimes B$. Then starting from equation (16) one has

$$
\begin{aligned}
e^{+\mathcal{L}_S t}\mathrm{Tr}_B\{(e^{-\mathcal{L}_S t}Ae^{+\mathcal{L}_S t} \otimes e^{-\mathcal{L}_B t}Be^{+\mathcal{L}_B t})(e^{-\mathcal{L}_S t'}Ae^{+\mathcal{L}_S t'} \otimes e^{-\mathcal{L}_B t'}Be^{+\mathcal{L}_B t'})\rho_B^e\}e^{-\mathcal{L}_S t}\sigma(t) = \\
\mathrm{Tr}_B\{(A \otimes e^{-\mathcal{L}_B t}Be^{+\mathcal{L}_B t})(e^{\mathcal{L}_S(t-t')}Ae^{-\mathcal{L}_S(t-t')} \otimes e^{-\mathcal{L}_B t'}Be^{+\mathcal{L}_B t'})\rho_B^e\}\sigma(t) = \\
\mathrm{Tr}_B\{(\tilde{A}(0) \otimes \tilde{B}(0))(\tilde{A}(t-t') \otimes \tilde{B}(t-t'))\rho_B^e\}\sigma(t) = \\
\mathrm{Tr}_B\{\tilde{\mathcal{L}}(0)\tilde{\mathcal{L}}(t-t')\rho_B^e\}\sigma(t),
\end{aligned}
\tag{3}
$$

which appears to be different from equation (19). Equation (20) on the other hands appears to be correct again.

To conclude, although this work approaches spin relaxation from an interesting angle, the paper overall appears as somewhat sloppy and rushed. I suggest to the authors to correct their misinterpretation regarding Malcom Levitts and my work, as well as some of the technical issues, before publication can be considered.

---

## Author Comment (AC3)

Please find below our response to the comments. We thank the reviewer for careful reading of the manuscript. We address the points raised by the referee below.
* * *
*The work by Rodin and Abergel revisits some of the early developments of spin relaxation theory in NMR and discusses their relation to the Lindblad equation, which is commonly employed in quantum optics and the mathematical physics literature. A detailed discussion on classical and quantum mechanical correlation functions/spectral densities with respect to detailed balance is also included.*

*1) The paper seems to be in response to a recent publication "A master equation for spin systems far from equilibrium" (doi.org/10.1016/j.jmr.2019.106645) authored by Malcolm Levitt and myself. However, several references and conclusions regarding our work are incorrect or misstated. In their abstract Rodin and Abergel claim that our work has "recently questioned" the "underlying theory" worked out by the founding fathers of spin relaxation theory. This is incorrect, we clearly state at the beginning of the article that our work explores the disadvantage and advantages of several thermalization techniques (IME, Jeener, Levitt-diBari) in the context of highly polarised spin systems (see section 4). In particular we show that the inhomogeneous master equation (IME) may lead to non-physical results for such systems. At no point do we cast doubt upon the underlying theory developed by Bloch, Wangness, Redfield, etc. On the contrary, we give full credit to the "founding fathers" already being "fully aware that semiclassical relaxation theory [...] is of limited validity" for highly polarised systems. If anything we cast doubt upon the current use of the IME for highly polarised spin systems within the NMR community.*

We thank the reviewer for these comments, and as we alredy stated in our reply to Malcolm Levitt's « community comment » on our manuscript, we realize that the wording used in our manuscript may have been unfortunate, when it comes to allude to your paper. We would therefore want to make it clear that your insightful « Bengs & Levitt » paper  points to the exact limitations of the semiclassical theory of relaxation (« high spin order », to be short). Although the assumptions that lead from the full quantum theory to this semiclassical formulation are applicable in many practical situations, the breakdown of this approximate theory and the way to overcome such limitations by using the Lindblad approach was clearly illustrated by several examples in your paper.
In our manuscript, we mention that your paper shows that the « formulation currently used by NMR spectroscopists », leads to erroneous predictions. This, of course, implicitly refers to the semiclassical theory. I hope this will dissipate any misunderstanding.
We thank the reviewer for a careful summary of his own paper. We have now modified the manuscript so as to make this point clearer.

*2) Within this context section 2.2 by Rodin and Abergel seems to stipulate that we imply that it is impossible to derive the Lindblad equation from the quantized theory of Bloch and Redfield. However, we simply presented the Lindblad method as a convenient approach to extend the semiclassical relaxation theory to systems outside the weak-order limit. We rather hoped that it would be clear that the Lindblad equation follows from the quantized BWR theory given that the starting point of our derivation in section 5 coincides with the starting point of BWR. However, it is true that we could have stated this fact in more detail as has recently been done by Tom Barbara ([https://doi.org/10.5194/mr-2-689-2021](https://doi.org/10.5194/mr-2-689-2021)).*

Reviewer #1 seems to overinterpret, or misinterpret our intentions, which we obviously need to clarify. In our manuscript, we wanted to show how the properties of the correlation functions emerge from the derivation of the master equation and how they relate to a semi-classical relaxation theory. Such properties are the consequences of both the Boltzmann equilibrium assumption for the lattice and of the non-commutation of the bath operators. Indeed, the Lindblad relaxation operator appears naturally, together with the associated spectral density functions that bear the important

asymmetry properties, from the perturbative solution of the evolution equation. These spectral properties and the associated Kubo relationships are direct consequences of the assumptions made on the lattice and lattice operators. Note in passing that the Kubo relations were originally defined for symmetrized correlation functions [J. Phys. Soc. Jpn (1957) 12, 570], whereas from the derivation of the master equation there emerge unsymmetrized correlation functions and spectral density functions (termed « left » and « right » in our manuscript).

The Lindblad equation makes no such assumption, and only assumes that spectral density functions reflect a Markovian process. It therefore requires additional adjustments to directly lead to a semiclassical theory. Indeed, because the bath operators $B^q$ do not commute, the correlation functions do not obey the general symmetry rules of classical correlation functions (see our Eq. 50). However, symmetrized correlation functions do, and so are introduced (Eq. 52). It is therefore the quantum mechanical master equation with these symmetrized (quantum-mechanical) correlation functions that should be used for a semi-classical theory (which we call « pseudo-classical » to distinguish it from the theory where the effect of the bath is taken into account only through random functions). We here recovered a result obtained by Hubbard.

Finally, we may have indeed misinterpreted misleading section titles in your paper, where paragraphs 4 and 5 are explicitly devoted to 'thermalization' issues, which apparently wrongly led us to believe that this was the focus. Incidently, we do not believe that « Lindblad thermalization » is adequate, as the purpose of the Lindblad approach is to provide a general form of Markovian quantum dissipative systems. There is no assumption as to the equilibrium state, the latter being assigned a particular form based on the additional, thermodynamical, hypothesis of the bath (lattice) being in a Boltzmann equilibrium. In this sense, there are more assumptions in the NMR relaxation theory than in the Lindblad approach, and the Bloch theory yields both the Kubo relations and the Lindblad equation in the special case of spin relaxation.

We have modified the manuscript so as to make this point clearer.

*3) The authors conclude section 2.2 with "the Lindblad equation [...] is thus derived from the usual quantized theory of relaxation." I am not sure if the authors are trying to claim novelty for this result (it also appears to be core statement of their abstract), if so they should consult the books by Breuer (DOI:10.1093/acprof:oso/9780199213900.001.0001), Schaller (10.1007/978-3-319-03877-3), and the recent article by Tom Barbara.*

We have assumed that this remark from reviewer #1 was a purely rethorical one, as our manuscript (including the title) insistently tends to demonstrate that, as Tom Barbara put it in a comment, « Bloch got it right from the beginning ».

*4) Unfortunately some mischaracterisation of our analysis of singlet-triplet relaxation in highly polarised spin systems may be found within the conclusion. The authors state: "The author's interpretation was that "thermalization" was not properly taken into account by the inhomogeneous semiclassical version of the Redfield equation, which was ascribed to the fact that the classical approach does not comply with the detailed balance condition." No such statements are made. The deviation between these results was not ascribed to detailed balance, but to the fact that the IME generates the unphysical situation in which singlet order may directly flow into Zeeman magnetisation (clearly illustrated in figure 1).*

We agree that this may have been an inaccurate interpretation of your paper. However, this is an interesting point that requires clarification, as the reviewer's response indicates. First, the term « classical » was meant to refer to the « conventional », semi-classical, approach, and indeed appears to be a bad choice.

The question is why the IME provides erroneous predictions in some cases. From the quantummechanical approach, it is obtained in the « low order » case, when the thermal corrections to the double commutator part of the relaxation operator are retained to first order in the largest eigenvalue $\mathfrak{H}_k/kT$, and the relaxation operator reduces to a double commutator. However, the equilibrium density operator is not a stationary solution in this case and therefore a correction term is added to the master equation and gives the same result as the usual semiclassical master equation. And so, when the low order assumption is not verified, as in the case of a spin system prepared in the singlet state, this description becomes inconsistent.

The non commutation of the bath operators have critical consequences, leading to the lattice temperature dependent terms in the master equation. Only when the bath operators $B^{-q}(t)$ and $B^q(0)$ commute, one recovers the double commutator expression, with the additional property $J_L(\mathfrak{H})=J_R(\mathfrak{H})$, so that the Kubo relation imposes an infinite lattice temperature. This illustrates how the finite temperature of the lattice is conveyed to the spins through non commutation of the bath operators of the coupling hamiltonian.

In the conventional semiclassical approach, the spin-bath interactions are represented by spin hamiltonians modulated by random functions of the lattice. Therefore, it has two simultaneous consequences : the structure of the relaxation operator is affected in such a way that the master equation takes the form of a double commutator ; and since $J_L(\mathfrak{H})=J_R(\mathfrak{H})$, the system cannot evolve to a thermodynamic equilibrium associated with a finite temperature. In this case, detailed balance property is conserved but only in the special case of infinite lattice temperature.

Thus, the ad hoc equilibrium density operator term used to account for a finite lattice temperature in the semi-classical theory, which makes the master equation inhomogeneous, can reflect the dynamics of the system only in situations where the non commuting terms can be neglected, which correspond, as stated by Redfield, to a situation where the spins are prepared in « an usual way ».

The manuscript has been modified to clarify this point.

*5) The provided discussion by the authors on this subject is also somewhat confusing. If I understand correctly the authors strongly emphasise the validity of detailed balance for the bath (in example the equilibrium distribution of the liquid follows Boltzmann). But this does not matter very much if the spin transitions do not obey detailed balance, which is clearly what our work refers to when talking about detailed balance. Consider for example equation (77) with a spectral density $J_\theta(\omega)=\theta(\omega)J(\omega)$, where $\theta(\omega)$ enforces detailed balance of the spectral densities. Consider now an ensemble of spin-1/2 particles experiencing random field relaxation in solution. According to equation (77) the relaxation superoperator is given by*

$$\hat{\Gamma} = -\frac{1}{2}\xi_{\mathrm{erf}}\sum_{m=-1}^{+1} J(|m|\omega)\theta(m\omega)\hat{T}_{1m}^{\dagger}\hat{T}_{1m}$$

*It is straightforward to verify the following matrix elements*

$$W_{\alpha\leftarrow\beta} = (P_\alpha|\hat{\Gamma}|P_\beta) = \frac{\xi_{\mathrm{erf}}}{4}J(\omega)(\theta(+\omega)+\theta(-\omega))$$

$$W_{\beta\leftarrow\alpha} = (P_\beta|\hat{\Gamma}|P_\alpha) = \frac{\xi_{\mathrm{erf}}}{4}J(\omega)(\theta(+\omega)+\theta(-\omega)),$$

*where $P_\alpha$ and $P_\beta$ are the $\alpha$ and $\beta$ population operators, respectively. The two transition rates per unit time $W_{\alpha\leftarrow\beta}$ and $W_{\beta\leftarrow\alpha}$ are equal. So even if the detailed balance condition holds for the bath it does not hold for the spin-state transitions when equation (77) is used. In other words it is correct to say that the equilibrium density operator for the spin system is not in the nullspace of $\Gamma$ˆ because it does not obey detailed balance for spin transitions. This has been discussed extensively in appendix A.2 of our work.*

This argument does not seem totally clear to us. You seem to be willing to demonstrate that the semiclassical theory of relaxation does not lead to detailed balance for the spins, and that there is no

connection between spin and bath detailed balance. My understanding is that the « eq. 77 » invoked here is the semi-classical relaxation operator. Quite obviously, one should not expect any connection between spin and bath thermal properties, as they are not taken into account. In addition, the non commutation of bath operators of the spin-bath coupling, which is central in the quantummechanical theory, as discussed above, is irrelevant in this case.

In contrast, we should point out that, in the full quantum theory, detailed balance for the spins is indeed conveyed from the detailed balance property of the bath (under the hypothesis of a bath in Boltzmann equilibrium).

6) *Besides the very unfortunate misinterpretation of our work, the provided discussion regarding classical and quantum mechanical correlation functions is interesting and is used to introduce a "thermalized double-commutator" in section 4. The thermalized double commutator renders the relaxation superoperator Redfield like and it is shown that this approach is fully equivalent to the Lindblad method. However advantages and disadvantages, if there are any, are not discussed. For example, the thermalized double-commutator does not represent a double-commutator in a strict mathematical sense and might be awkward to incorporate into existing simulation software.*

We would agree that the term « thermalized double commutator » may lack elegance, but it reflects the underlying considerations developed in our manuscript and in this letter. The spectral density functions involved obey general symmetry properties of classical correlation functions , which can therefore be used in a straigthforward manner to generate semi-classical approximations of the master equation.

7) *Additionally, representing the relaxation superoperator in Lindblad form does not only provide a simple check on positivity and Markov compatibility, but is also in agreement with the majority of the literature outside NMR.*

The formulation used in our manuscript is totally in line with Lindblad. This has been explicitly indicated in several instances (see paragraph 3 of our manuscript, in particular. ). The point that seemed important to make regarding the Lindblad formulation is that, since it says nothing about the thermodynamic equilibrium of the bath, such assumptions must be introduced explicitly in a way that is not directly connected to the physical problem at hand, in contrast to the Bloch-Redfield approach, from which critical properties of the spin correlation functions can be deduced.

8) *Apart from that there also some technical issues.*

• *The trace operator Tr should not be italicized.*
This has been corrected
• *The interaction frame transformation above equation (4) should read as* $\exp\left(-L\square_0 t\right)$
This has been corrected
• *The authors switch to a time dierence variable* $\tau = t - t'$ *in equation (6), but this is not consistent with equations (7)-(19).*

It is true that the notation seems inconsistent. We simply switched to the (t, tau) variables for final expressions, and kept (t,t') for the intermediate steps. However, after Eq 33, we use (t,tau) for the calculation of the bath correlation functions so as to emphasize their stationarity property.

• *Something appears to be wrong with equation (19), take for example* $L_1 = A \otimes B$
*Then starting from equation (16) one has*

$$e^{+\mathcal{L}_S t}\mathrm{Tr}_B\{(e^{-\mathcal{L}_S t}Ae^{+\mathcal{L}_S t} \otimes e^{-\mathcal{L}_B t}Be^{+\mathcal{L}_B t})(e^{-\mathcal{L}_S t'}Ae^{+\mathcal{L}_S t'} \otimes e^{-\mathcal{L}_B t'}Be^{+\mathcal{L}_B t'})\rho_B^e\}e^{-\mathcal{L}_S t}\sigma(t) =$$

$$\mathrm{Tr}_B\{(A \otimes e^{-\mathcal{L}_B t}Be^{+\mathcal{L}_B t})(e^{\mathcal{L}_S(t-t')}Ae^{-\mathcal{L}_S(t-t')} \otimes e^{-\mathcal{L}_B t'}Be^{+\mathcal{L}_B t'})\rho_B^e\}\sigma(t) =$$

$$\mathrm{Tr}_B\{(\tilde{A}(0) \otimes \tilde{B}(0))(\tilde{A}(t-t') \otimes \tilde{B}(t-t'))\rho_B^e\}\sigma(t) =$$

$$\mathrm{Tr}_B\{\tilde{\mathcal{L}}(0)\tilde{\mathcal{L}}(t-t')\rho_B^e\}\sigma(t),$$

Notation inconsistencies in the Schrödinger representation have been corrected and the complete derivation is given in an Appendix.

---

## Author Comment (AC4)

*This paper certainly has generated already considerable discussion and I do not find much to add that has not already been said elegantly by other reviewers and commentators. It is often the case in science that alternative derivations of the same result provide new insights into the underlying physical principles, and I think that this paper, following on recent papers by Bengs and Levitt and Barbara, is a good example of the phenomenon. I found the discussion around the left and right spectral density functions and the interrelationship to Boltzmann factors to be particularly interesting and I suspect will be interesting to the many students of the subject who are, like this reviewer, more familiar with classical spectral density functions*

We wish to thank the reviewer for his comments and for finding interest in our work.

---

## Author Comment (AC5)

*This paper, together with a recent paper by Bengs and Levitt, discusses and revisits the problem of the relaxational dynamics of a collection of interacting spins coupled to the vibrations of the lattice. It is common to assume that the spins are weakly coupled to the lattice and to use a perturbative approach (Born-Markov approximation) that allows to write a master equation of the Redfield kind for the density matrix reduced to the spin degrees of freedom [equation 21 of the main text]. Equation 21 does not have a Lindblad form. A Lindblad form is required to respect the physical properties of a density matrix (positive, trace conserving…).*

*In practice, in the literature further approximations are proposed for the evolution equation of the spin density matrix. The first two (i.e. the ones discussed in this paper) are:*

*To introduce a secular approximation that reduces Eq. 21 to Eq 40 or the identical Eq. 43. The equation has a Lindblad form and, by construction, relaxes the interacting spins to their Boltzmann equilibrium. Namely to \sigma_{Boltzmann) = exp(-\beta H_S)*

*In NMR (however see the historical paper by Tom Barbara) one is used instead to a semi-classical approximation of the Redfield equation which leads to Equation 65. Unfortunately Eq.65 does not have a Lindblad form. In general, for more than 1 spin this equation leads to misleading results that are discussed in this paper as well as in the previous paper by Bengs and Levitt. In my opinion this equation is very clumsy: on one side it does not reproduce a physical evolution and one the other side it is phenomenological. It works when the physics of the problem is well described by a single spin, but it will fail to capture many body effects, among others. Still within the Markovian assumption, instead of designing further approximations to turn Redfield's equation into a physically sound form (Lindblad), one can go one step back and examine the many possible ways of making the Markovian approximation. This leads to another proposal I would like to mention:*

 *Recently a more general Lindblad evolution has been proposed (PERLind approaches, see e.g. Nathan & Rudner PRB 2020). It holds at the same level of approximation as Redfield's (Eq.21), i.e. second-order perturbation with respect to the coupling to the lattice, but it is of Lindblad form. In the limit of very weak coupling with the lattice one recovers the secular approximation of Eq. 40. For moderate coupling it captures the competition between the interactions among the spins and the coupling with the lattice. As a result the stationary state of this equation is not exactly exp(-\beta H_S) even if, in a strong magnetic field, the total magnetisation will be indistinguishable from the one predicted by Boltzmann. Indeed Boltzmann equilibrium is not expected to hold outside weak system-bath coupling. Recently we used this arguably more general equation to show that the spin temperature (generated by dipolar interactions) can be suppressed at high temperature due to the effect of the lattice vibrations (Maimbourg, Basko, Holtzmann, Rosso, PRL 2021).*

*In conclusion I think it is important (within the Markovian assumption) to stick with well-defined Lindblad forms and I think that this discussion is important. I also wish to advertise that a lot of physics can be found beyond the secular approximation.*

We would like to thank you for the insightful comments and for pointing out recent investigations of Linblad operators. We added the suggested references to the papers of Nathan & Rudner, and  of Maimbourg et al. on generalizations of the Linblad approach in the revised manuscript.

---

## Author Response (AR2)

Daniel Abergel, MD, PhD

Laboratoire des biomolécules
Ecole Normale Supérieure,
Département de Chimie,
24 rue Lhomond, 75231 Paris cedex 05, France          Tel : (+33) 1 44 32 32 65
Email: daniel.abergel@ens.psl.eu

Paris, January 3, 2022

Dear Editor,

Please find below our reply to the comments of the reviewers.
We wish to thank all reviewers for their careful reading and their constructive comments on this manuscript.

Besides, I should mention that we have added funding details in the « Financial Support » section that must appear in the published version of the manuscript.
Sincerely,

Daniel Abergel

**Response to Reviewer #5**

Please find below our response to the comments. We thank the reviewer for careful reading of the manuscript.

*It has been recently shown that the "thermalized" semi-classical approach used in conventional treatments of NMR relaxation, while reliable in the treatment of "standard" experiments become unreliable in systems, such as hyperpolarized states, that are far from equilibrium. Treatment of such open quantum-mechanical systems is generally carried out in the formalism of a Lindblad master equation. The authors Rodin and Abergel show that this is formally equivalent to the standard 2nd order perturbation approach within a fully quantum mechanical framework under the usual approximations used in the treatment of NMR relaxation. I feel that this work is important in marrying the conventional perturbation approach that most NMR spectroscopists are familiar with and the "more formal" approaches to open quantum systems. I feel that this manuscript serves as an important (and complete) pedagogical framework for students of NMR relaxation to appreciate both the mathematical tools and the concepts encoded therein. I congratulate the authors for having produced such a thorough analysis.*

*I have only a few (very) minor suggestions –*
*1. I assume that the title implies that the manuscript integrates all approaches to NMR relaxation "under the sun", not sure that it is formally true though I agree that the authors marry two (major) seemingly orthogonal approaches.*

Actually, this title was not meant to refer to some extensive review of the many formulations of relaxation theory. We just addressed some fundamental aspects of the quantum mechanical formulation of the problem to emphasize the fact that all these aspects have been identified, if not treated, in the early days of NMR. Thus, the question mark in the title is mostly a rethorical one, as we fear that the answer is negative, although we are extremely pleased to see that it has aroused some interest.

*2. Starting with (2), the equations are written in units of hbar. Therefore, the appearance of hbar in the Boltzmann terms e.g., in (36) makes things somewhat inconsistent.*

The indication that ℏ =1 has been added to the text.

*3. Define L in (36) for completeness.*

The definition of L as the trace of $\exp(\beta\, H_B)$ has been added to the manuscript.

*4. For (11), the authors should use the appropriate direct product symbol.*

Corrected

*5. In (23) perhaps it is worth stating that the expansion is in the basis of irreducible tensor operators of rank q; though not absolutely necessary, this is what is generally done for NMR.*

We prefer to leave this point undefined, as it involves subtelties of notations that may obscure the demonstration and be misleading at some point.

*6. There are several instances where it is said that a certain equation "writes" e.g., before (26). These occurrences should be replaced by "becomes".*

Done

*7. After (20), I would replace "…these denote functions…" by "…these denote standard time-correlation functions rather than….".*

Done

*8. After (45), I would replace the phrase ".. express a Kubo kind of relation.." by "…express a relation similar to those provided by Kubo …"*

*9. Line 305, define trace as Tr as in other cases.*

Done

*10. The Goldman classic "Formal Theory …." is from 2001 not 2021.*

We warmly thank the reviewer for pointing out this unfortunate typo.
!